# Sparse Coding with Gated Learned ISTA

**Kailun Wu** [*†]
Department of Automation,
Tsinghua University
Beijing, P.R.China
wukl14@mails.tsinghua.edu.cn
&kailun.wukailun@alibaba-inc.com

**Yiwen Guo** [*]
Bytedance AI Lab
Beijing, P.R.China
guoyiwen.ai@bytedance.com

**Ziang Li** [†]
Department of Automation,
Tsinghua University
Beijing, P.R.China
liza19@mails.tsinghua.edu.cn

**Changshui Zhang** [†]
Department of Automation,
Tsinghua University
Beijing, P.R.China
zcs@mail.tsinghua.edu.cn

## Abstract

In this paper, we study the learned iterative shrinkage thresholding algorithm (LISTA) for solving sparse coding problems. Following assumptions made by prior works, we first discover that the code components in its estimations may be lower than expected, i.e., require gains, and to address this problem, a gated mechanism amenable to theoretical analysis is then introduced. Specific design of the gates is inspired by convergence analyses of the mechanism and hence its effectiveness can be formally guaranteed. In addition to the gain gates, we further introduce overshoot gates for compensating insufficient step size in LISTA. Extensive empirical results confirm our theoretical findings and verify the effectiveness of our method.

## 1 Introduction

Sparse coding serves as the foundation of many machine learning applications, e.g., the direction-of-arrival estimation (Xu et al., 2012), signal denoising (Elad & Aharon, 2006), and super resolution imaging (Yang et al., 2010). In general, it aims to recover an inherently sparse vector $x_s \in \mathbb{R}^n$ from an observation $y \in \mathbb{R}^m$ corrupted by a noise vector $\varepsilon \in \mathbb{R}^m$. That is,

$$y = Ax_s + \varepsilon, \tag{1}$$

in which $A \in \mathbb{R}^{m \times n}$ is an over-complete basis matrix. The problem of recovering $x_s$, however, is a challenging task, in which the main difficulties are to incorporate the sparse constraint which is nonconvex and to further determine the indices of its non-zero elements, i.e., the support of the vector. A reasonable solution to the problem is to use convex functions as surrogates to relax the constraint of sparsity, among which the most classical one probably is the $l_1$-norm penalty. Such a problem is carefully studied in Lasso (Tibshirani, 1996), and it can be solved via least angle regression (Efron et al., 2004), the iterative shrinkage and thresholding algorithm (ISTA) (Daubechies et al., 2004), etc.

Despite the simplicity, these conventional solvers suffer from critical shortcomings. Taking ISTA as an example, we know that 1) it converges very slowly with only a sublinear rate (Beck & Teboulle, 2009), 2) the correlation between each of the two columns of $A$ should be relatively low. In recent years, deep learning (LeCun et al., 2015) methods have achieved remarkable successes. Deep neural

---

[*]Kailun Wu and Yiwen Guo contribute equally to the paper.

[†]Kailun Wu, Ziang Li, and Changshui Zhang are with the Institute for Artificial Intelligence (THUAI), the State Key Lab of Intelligent Technologies and Systems, the Beijing National Research Center for Information Science and Technology (BNRist), and the Department of Automation, Tsinghua University. Kailun Wu did his work when he was a Ph.D. student at Tsinghua Unversity, and now he works at Alibaba Group. This work is (jointly or partly) funded by the NSFC (Grant No. 61876095) and Beijing Academy of Artificial Intelligence (BAAI).

networks (DNNs) have been proven both effective and efficient in dealing with many tasks, including image classification (He et al., 2016), object detection (Girshick, 2015), speech recognition (Hinton et al., 2012), and also sparse coding (Gregor & LeCun, 2010; Wang et al., 2016; Borgerding et al., 2017; He et al., 2017; Zhang & Ghanem, 2018; Chen et al., 2018; Liu et al., 2019; Sulam et al., 2019). The core idea behind deep learning-based sparse coding is to train DNNs to approximate the optimal sparse code. For instance, an initial work of Gregor and LeCun's (2010) takes the inspiration from ISTA and develops an approximator named learned ISTA (LISTA), which is structurally similar to a recurrent neural network (RNN).

It has been demonstrated both empirically and theoretically that LISTA is superior to ISTA (Wang et al., 2016; Moreau & Bruna, 2017; Giryes et al., 2018; Chen et al., 2018). Nevertheless, it is also uncontroversial that there exists much room for further enhancing it. In this paper, we delve deeply into the foundation of (L)ISTA and discover possible weaknesses of LISTA. First and foremost, we know from prior arts (Chen et al., 2018; Liu et al., 2019) that LISTA tends to learn large enough biases to achieve no "false positive" in the support of generated codes and further ensure linear convergence, and we prove that this tendency, however, also makes the magnitude of the code components being lower than that of the ground-truth. That said, there probably exists a requirement of *gains* in the code estimations. Second, regarding the optimization procedure of ISTA as to minimize an upper bound of its objective function at each step, we conjecture that the element-wise update of (L)ISTA normally "lags behind" the optimal solution, which suggests that it requires *overshoots* to reach the optimum, just like what has been suggested in fast ISTA (FISTA) (Beck & Teboulle, 2009) and learned FISTA (LFISTA) (Moreau & Bruna, 2017).

In this paper, our main contributions are summarized as follows:

- We discover weaknesses of LISTA by theoretically analyzing its optimization procedure, for mitigating which we introduce gain gates and overshoot gates, akin to update gate and reset gate mechanisms in the gated recurrent unit (GRU) Cho et al. (2014).

- We provide convergence analyses for LISTA (with or without gates), which further give rise to conditions on which the performance of our method with gain gates can be guaranteed. A practical case is considered, where the assumption of no "false positive" is relaxed.

- Insightful expressions for the gates are presented. In comparison with state-of-the-art sparse coding networks (not limited to previous extensions to LISTA), our method achieves superior performance. It also applies to variants of LISTA, e.g., LFSITA (Moreau & Bruna, 2017) and ALISTA (Liu et al., 2019).

**Notations**: In this paper, unless otherwise clarified, vectors and matrices are denoted by lowercase and uppercase characters, respectively. For vectors/matrices originally introduced without any subscript, adding a subscript (e.g., $i$) indicates its element/column at the corresponding position. For instance, for $x \in \mathbb{R}^n$, $x_i$ represents the $i$-th element of the vector, and $W_{:,i}$ and $W_{i,:}$ denote the $i$-th column and row of a matrix $W$ respectively. While for vectors introduced with subscripts already, e.g., $x_s$, we use $(x_s)_i$ to denote its $i$-th element. The operator $\odot$ is used to indicate element-wise multiplication of two vectors. The support of a vector is denoted as $\mathrm{supp}(x) := \{i | x_i \neq 0\}$. We use $\sup_{x_s}$ as the simplified form of $\sup_{x_s \in \mathcal{X}(B,s,0)}$, see Assumption 1 for the definition of $\mathcal{X}(B,s,0)$.

## 2 BACKGROUND

In general, sparse coding solves the problem that can be formulated as

$$\min_x f(x,y) + \lambda r(x), \tag{2}$$

in which $f(x,y)$ calculates the residual of approximating $y$ using a linear combination of column-wise features in $A$. The function $f(x,y)$ is convex with respect to $x$ in general. In particular, if $\varepsilon$ is a Gaussian vector, then it should be $f(x,y) = \|Ax - y\|_2^2$. The term $\lambda r(x)$ serves as a regularizer for sparsity and we have $r(x) = \|x\|_1$ in Lasso. As mentioned, a variety of algorithms can be applied to solve the problem and our focus in the paper is (L)ISTA. We first revisit the optimization procedure of ISTA, which is the foundation of LISTA as well. Given $y$, let us introduce a scalar $\gamma > 0$ that fulfills $\gamma I - \nabla_x^2 f(x,y) \succ 0, \forall x$, then it can be considered as optimizing an upper bound of the objective

function obtained via Taylor expansion. To be more specific, for any presumed $x^{(t)}$, we have

$$f(x, y) + \lambda r(x) \leq f(x^{(t)}, y) + (x - x^{(t)}) \nabla_x f(x^{(t)}) + \frac{\gamma}{2} \|x - x^{(t)}\|^2 + \lambda r(x). \qquad (3)$$

By substituting $r(x)$ with $\|x\|_1$ and optimizing the bound in an element-wise manner, we can easily get the one-step update rule that zeros the gradient based on $x^{(t)}$. It is, $x^{(0)} = \mathbf{0}$ and

$$x^{(t+1)} = s_{\lambda/\gamma}(x^{(t)} - \nabla_x f(x^{(t)})/\gamma), \quad \forall t \geq 0, \qquad (4)$$

in which $s_b(x) := \text{sign}(x)(|x| - b)_+$ is a shrinking function and $(\cdot)_+$ is a rectified linear unit (ReLU) calculating $\max\{0, \cdot\}$. For Gaussian noises, the formulation reduces to

$$x^{(t+1)} = s_{\lambda/\gamma}\left(\left(I - \frac{A^T A}{\gamma}\right) x^{(t)} + \frac{A^T}{\gamma} y\right). \qquad (5)$$

The update as shown in Eq. (4) and (5) can be performed iteratively until convergence. However, the convergence of ISTA (along with some other conventional solvers) is known to be slow, and it has been shown that DNNs can be utilized to accelerate the procedure. Many researchers have explored the idea since the initial work of Gregor and LeCun's (i.e., LISTA). For LISTA, they design deep architectures following the main procedure of ISTA yet to learn parameters in an end-to-end manner from data (Gregor & LeCun, 2010; Hershey et al., 2014). The inference process of LISTA is similar to that of an RNN and can be formulated as $x^{(0)} = \mathbf{0}$ and

$$x^{(t+1)} = s_{b^{(t)}}(W^{(t)} x^{(t)} + U^{(t)} y), \quad t = 0, \cdots, d - 1, \qquad (6)$$

where $\Theta = \{U^{(t)}, W^{(t)}, b^{(t)}\}_{t=0,1,\ldots,d-1}$, is learnable parameters set. Some works (Xin et al., 2016; Chen et al., 2018) have proved that $W^{(t)}$ and $U^{(t)}$ should satisfy the constraint $W^{(t)} = I - U^{(t)} A$, such that

$$x^{(t+1)} = s_{b^{(t)}}(x^{(t)} + U^{(t)}(Ax^{(t)} - y)), \quad t = 0, \cdots, d - 1. \qquad (7)$$

The parameters in $\Theta$ are normally learned from a set of training samples by minimizing the difference between the final code estimations and ground-truth. In this paper, our main assumption for theoretical analyses follows those of prior works (Chen et al., 2018; Liu et al., 2019) in a noiseless case, and noisy cases will be considered in the experiments.

**Assumption 1.** *The sparse vector $x_s$ and noise vector $\varepsilon$ are sampled from a set $\mathcal{X}(B, s, 0)$ fulfilling:*

$$\mathcal{X}(B, s, 0) := \{x \big| \|x\|_\infty \leq B, \varepsilon = \mathbf{0}, \|x\|_0 \leq s\}.$$

## 3 SPARSE CODING WITH GAIN GATES AND OVERSHOOT GATES

In this section, we will introduce the advocated gain gates and overshoot gates. Along with thorough discussions for the motivations, their formulations are provided in Section 3.1 and 3.2, respectively. Figure 1 summarizes the inference process of the standard LISTA and two evolved versions with our gates incorporated. Proofs of all our theoretical results are deferred to the appendix.

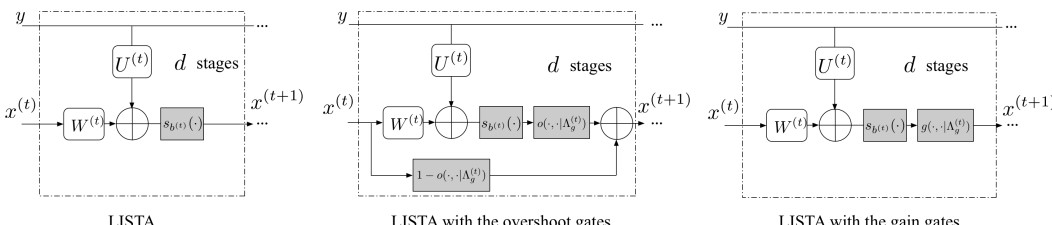

Figure 1: The inference process of the standard LISTA and evolved versions with our gates

### 3.1 SPARSE CODING WITH GAIN GATES

Recent works have shown linear convergence of LISTA (Chen et al., 2018; Liu et al., 2019). In order to guarantee the convergence, it is also demonstrated that the value of bias terms should be large enough to eliminate all "false positive" in the support of the generated codes. However, this may lead to an issue that the magnitude of the generated code components in LISTA must be smaller than or at most equal to those of the ground-truth. Our result in Proposition 1 makes this formal. For clarity of the result, we would like to introduce the following definition first.

**Definition 1.** *(Liu et al., 2019) Given a matrix $A \in \mathbb{R}^{m \times n}$, its generalized mutual coherence is:*

$$\mu(A) := \inf_{W \in \mathbb{R}^{n \times m}, W_{i,:} A_{:,i}=1, \forall i} \left\{ \max_{i \neq j, 1 \leq i,j \leq n} W_{i,:} A_{:,j} \right\}. \tag{8}$$

*We let $\mathcal{W}(A)$ denote a set of all matrices that can achieve the generalized mutual coherence $\mu(A)$, which means:*

$$\mathcal{W}(A) := \left\{ W \big| \max_{i \neq j, 1 \leq i,j \leq n} W_{i,:} A_{:,j} = \mu(A), W_{i,:} A_{:,i} = 1, \forall i \right\}. \tag{9}$$

**Proposition 1.** *(Requirement of gains). With $U^{(t)} \in \mathcal{W}(A)$ and $W^{(t)} = I - U^{(t)}A$, if $b^{(t)} = \mu(A) \sup_{x_s} \|x^{(t)} - x_s\|_1$ is achieved in LISTA to guarantee no "false positive" (i.e., $supp(x^{(t)}) \subset supp(x_s)$) and further linear convergence (i.e., $\|x^{(t)} - x_s\|_2 \leq sB \exp(ct)$, in which $c = \log((2s - 1)\mu(A))$), then we have for the estimation $|x_i^{(t)}| \leq |(x_s)_i|$ and $x_i^{(t)}(x_s)_i \geq 0, \forall i \in supp(x_s)$.*

Provided Proposition 1 as the evidence of a potential weakness of LISTA, we believe that if the code components can be enlarged appropriately, then the estimation at each step would be closer to $x_s$, and the convergence of LISTA will be further improved, which inspires us to design a gate to enlarge the generated code components. Such a gate is named as a gain gate and it acts on the input to the current estimation, akin to a reset gate in GRU (Cho et al., 2014), which is

$$x^{(t+1)} = s_{b^{(t)}}(W^{(t)}(g_t(x^{(t)}, y|\Lambda_g^{(t)}) \odot x^{(t)}) + U^{(t)}y), \tag{10}$$

in which the gate function $g_t(\cdot, \cdot|\Lambda_g^{(t)})$ outputs an $n$-dimensions vector, and $\Lambda_g^{(t)}$ is the set of its learnable parameters. In the original implementation of LISTA, the output of each layer is obtained by calculating Eq. (4) iteratively. It has been proven that the estimation $x^{(t)}$ ultimately converges to the ground-truth $x_s$ (as $t \to \infty$), only if the condition of $(W^{(t)} - (I - U^{(t)}A)) \to 0$ holds. That said, it is suggested that $U^{(t)}$ and $W^{(t)}$ are entangled to the end. Yet, with our gated mechanism, the update rule in neural networks has been modified into Eq. (10), making it unclear whether the convergence is guaranteed similarly or not. To figure it out, we perform theoretical analyses in depth, which will further provide guidance for the gate design. We are going to explore: whether the learnable matrices are still entangled as in LISTA, and to encourage fast convergence, what properties should the gate function satisfy? Theorem 1 and 2 give some answers to these questions and they are based on the same assumptions as for Proposition 1.

**Theorem 1.** *If the $s$-th principal minor of $W^{(t)}$ have full rank, then for the gate function bounded from both above and below, we have $x_s$ as the fixed point of Eq. (10) only if*

$$\text{diag}(g_t(x^{(t)}, y|\Lambda_g^{(t)})) \to D \quad and \quad W^{(t)}D - (I - U^{(t)}A) \to 0, \quad as\ t \to \infty, \tag{11}$$

*in which $D$ is an $n \times n$ constant diagonal matrix and the function $\text{diag}(\cdot)$ creates a diagonal matrix with the elements of its input on the main diagonal.*

From Theorem 1 we can equivalently have $(\tilde{W}^{(t)} - (I - U^{(t)}A)) \to 0$ by defining $\tilde{W}^{(t)} := W^{(t)}D$, which means the learnable matrices are similarly entangled as in the standard LISTA. Besides, we know that as the number of layers increases, each introduced gain gate should ultimately converge to a constant (diagonal) matrix $D$ to guarantee performance. Then if $W^{(t)} \to I - U^{(t)}A$, the gain gate function converges to an identical mapping as $t \to \infty$, and vice versa. This inspires us to "split" the gate function into an identical one and a residual one, and we thus advocate, for each index $i$ of the vector, the $i$-th element of gain gate is

$$g_t(x^{(t)}, y|\Lambda_g^{(t)})_i = 1 + \kappa_t(x^{(t)}, y|\Lambda_g^{(t)})_i \quad and \quad \kappa_t(x^{(t)}, y|\Lambda_g^{(t)})_i \geq 0, \tag{12}$$

in which $\kappa_t(x^{(t)}, y|\Lambda_g^{(t)})_i$ is the $i$-th element of $\kappa_t(x^{(t)}, y|\Lambda_g^{(t)})$, and it should decrease as $t$ increases, in order to guarantee convergence in Eq. (11). Let us further study the convergence rate of "LISTA" equipped with such gain gates. For clarity, we introduce another condition for the function before moving to more details:

$$\kappa_t(x^{(t)}, y|\Lambda_g^{(t)})_i < 2b_i^{(t-1)}/|x_i^{(t)}|. \tag{13}$$

We present theoretical results as follows on the basis of Proposition 1, i.e., we still have $U^{(t)} \in \mathcal{W}(A)$, $W^{(t)} = I - U^{(t)}A$ and Assumption 1, but the requirement for $b^{(t)}$ is different.

**Theorem 2.** *If $b^{(t)} = \mu(A) \sup_{x_s} \|x_s - x^{(t)} \odot g_t(x^{(t)}, y|\Lambda_g^{(t)})\|_1$ is achieved, following the update rule in Eq. (10), if the conditions in Eq. (12) and (13) hold for the gate function, there will be*

$$\|x^{(t)} - x_s\|_2 \leq sB \exp(\sum_{i=1}^{t-1} c_i + c), \tag{14}$$

*in which $c = \log((2s-1)\mu(A))$, $c_i = c$ if $i \leq \lceil \log(\frac{sB}{\|x_s\|_1}) / \log(\frac{1}{(2s-1)\mu(A)}) \rceil$, and $c_i < c$ otherwise.*

Theorem 2 presents an upper bound of $\|x^{(t)} - x_s\|_2$ for LISTA with gain gates, and it shows that so long as the gates satisfying conditions in Eq. (12) and (13) are introduced, the "convergence factor" $c + \sum c_i$ of our gated LISTA would be smaller in comparison with that of the standard LISTA (which is $ct$, see Proposition 1 and Chen et al.'s work 2018).

By consolidating all these theoretical cues, we further give principled expressions for the gate function. One may expect to endow the gates some learning capacities, thus we let

$$g_t(x^{(t)}, y|\Lambda_g^{(t)}) = 1 + \kappa_t(x^{(t)}, y|\Lambda_g^{(t)}) = 1 + \mu_t b^{(t-1)} f_t(x^{(t)}|\nu_t), \tag{15}$$

in which $\mu_t \in \mathbb{R}$ is a parameter to be learned, $b^{(t-1)}$ is the threshold parameter of the $(t-1)$-th layer, and $f_t(x^{(t)}|\nu_t)$ is a newly introduced function constrained not to be greater than $2/|x^{(t)}|$. We are going to evaluate different choices for the function $f_t(x^{(t)})$ in experiments, e.g.,

$$\text{the piece-wise linear function:} \quad f_t(x^{(t)}|\nu_t) = \text{ReLU}(1 - \text{ReLU}(\nu_t|x^{(t)}|)),$$

$$\text{the inverse proportional function:} \quad f_t(x^{(t)}|\nu_t) = 1/(\nu_t|x^{(t)}| + \epsilon), \tag{16}$$

$$\text{the exponential function:} \quad f_t(x^{(t)}|\nu_t) = \exp(-\nu_t|x^{(t)}|),$$

in which $\nu_t \in \mathbb{R}$ is a parameter to be learned, and $\epsilon$ is a tiny positive scalar introduced to avoid zero being divided. All the learnable parameters in a gain gate are thus collected as $\Lambda_g^{(t)} = \{\mu_t, \nu_t\}$.

### 3.1.1 NO FALSE POSITIVE?

Our previous theoretical results show that the performance of LISTA can be improved by using a gain gate, as long as the gate function satisfies conditions in Eq. (12) and (13), and no "false positive" is encountered. However, it is not always true in practice. Our experimental results also show that when the inverse proportional function is adopted as gain gates in lower layer for LISTA, the performance of our gated LISTA may even degrade. We conjecture that such contradiction to the theoretical results may be owing to impractical assumptions. In this subsection, we try to relax the assumption about no "false positive", and we further found that a tighter bound can be achieved with a more reasonable assumption instead. Through theoretical analyses as follows, we also demonstrate that the inverse proportional gain function should better be only adopted in higher layers. For clarity of the results, we would like to introduce the following definition first.

**Definition 2.** *Given a model with $\Theta$, in which $b^{(t)} = \Gamma \mu(A) \sup_{x_s} \|x^{(t)} \odot g_t(x^{(t)}, y|\Lambda_g^{(t)}) - x_s\|_1$, we introduce $\omega_{t+1}(k|\Theta)$ to characterize its relationship with the false positive rate, which is*

$$\omega_{t+1}(k_{t+1}|\Theta) = \sup_{\forall x_s, |supp(\check{x}^{(t+1)}) \cup supp(x_s)| \leq |supp(x_s)| + k_{t+1}} \Gamma,$$

*in which $\check{x}^{(t+1)} := s_{b^{(t)}}(W^{(t)}(x^{(t)} \odot g_t(x^{(t)}, y|\Lambda_g^{(t)}) - x_s))$, and $k_{t+1} \geq 0$ is the desired maximal number of "false positive" of $x^{(t+1)}$.*

The above definition applies to both the standard LISTA and LISTA with gain gates (we can let the gate function be an identity function to achieve a standard LISTA). We first analyze the convergence of LISTA without gates. We present theoretical results as follows on the basis of similar assumptions (including Assumption 1, $U^{(t)} \in \mathcal{W}(A)$, and $W^{(t)} = I - U^{(t)}A$), but with a different requirement for $b^{(t)}$ from Proposition 1.

**Theorem 3.** *If $b^{(t)} = \omega_{t+1}(k_{t+1}|\Theta)\mu(A) \sup_{x_s} \|x^{(t)} - x_s\|_1$ is achieved, and $\exists 0 < k_0^{(t)} < s$ such that $\omega_t(k_0^{(t)}|\Theta) < 1 - 1/(s - k_0^{(t)})$, then there exists "false positive" with $0 < k_t < s$ and*

$$\|x^{(t)} - x_s\|_2 \leq sB \exp(\sum_{i=1}^{t} c_i^*),$$

*in which $c_i^* < \log((2s-1)\mu(A))$.*

It can be seen that when we relax the assumption about no "false positive" and further reduce the value of the threshold $b^{(t)}$, the error bound of LISTA becomes even lower. Obviously, the previous bound of LISTA with gain gates in Theorem 2 is not necessarily lower than the tighter bound of a standard LISTA in Theorem 3, which well explains the contradiction of theoretical and empirical results. Here we re-deduce the error bound of our gated LISTA with the inverse proportional function in the following theorem. Note that we still have $U^{(t)} \in \mathcal{W}(A)$, $W^{(t)} = I - U^{(t)}A$ and Assumption 1.

**Theorem 4.** *Suppose that* $\min_{i \in supp(x_s)} |(x_s)_i| \geq \sigma > 0$, *if* $b^{(t)} = \omega_{t+1}(k_{t+1}|\Theta)\mu(A) \sup_{x_s} \|x^{(t)} \odot g_t(x^{(t)}, y|\Lambda_g^{(t)}) - x_s\|_1$ *is achieved and* $\exists 0 < k_0^{(t)} < s$ *such that* $\omega_t(k_0^{(t)}|\Theta) < 1 - 1/(s - k_0^{(t)})$, *then*

$$\|x^{(t)} - x_s\|_2 \leq sB \exp(\sum_{i=1}^{t-1} c_i' + c_t'^*),$$

*in which* $c_t'^* < \log((2s-1)\mu(A))$. $\exists t_0 = \lceil \log(\frac{sB}{\sigma}) / \log(\frac{1}{(2s-1)\mu(A)}) \rceil$ *if the scaling factor* $\mu_i$ *of the gate has* $\mu_i = 0$ *for* $i \leq t_0$, $0 < k_i < s$, *then* $c_i' = c_i^*$, *and if* $1 - \omega_i(s|\Theta) < \mu_i \leq 1$ *for* $i > t_0$, $k_i = 0$, *then* $c_i' < c_i^*$.

We can conclude from Theorem 4 that, a) a gain gate expressed by the inverse proportional function should be applied to deeper layers in LISTA, rather than lower layers, b) when using the function, there indeed exists no "false positive" (i.e., $k_i = 0$) in deeper layer. We follow such guidelines in the implementation of our gated LISTA. In addition, we observe that unlike the inverse proportional function, other considered functions show consistent performance gains on both lower and higher layers, hence we attempt to utilize them on lower layers in alliance with the inverse proportional function powered gain gates on the other layers. In practice, we choose the ReLU-based piece-wise linear function, and it is uniformly applied to the first 10 layers. We will empirically compare different choices between the gain gate functions in Section 4.1.

## 3.2 SPARSE CODING WITH OVERSHOOT GATES

Unlike the gain gates that are incorporated before performing estimation at each step, the overshoot gates act more like adjustments to the outputs, which can be viewed as learnable boosts:

$$\begin{aligned} \tilde{x}^{(t+1)} &= s_{b^{(t)}}(W^{(t)}x^{(t)} + U^{(t)}y), \\ x^{(t+1)} &= o_t(x^{(t)}, y|\Lambda_o^{(t)}) \odot \tilde{x}^{(t+1)} + (1 - o_t(x^{(t)}, y|\Lambda_o^{(t)})) \odot x^{(t)}. \end{aligned} \quad (17)$$

The gate function $o_t(\cdot, \cdot|\Lambda_o^{(t)}) : \{\mathbb{R}^n, \mathbb{R}^m\} \to \mathbb{R}^n$ outputs an $n$-dimensional vector and $\Lambda_o^{(t)}$ collects all the trainable parameters in the function, akin to a dedicated update GRU gate (Cho et al., 2014).

Our motivation comes from analyses of ISTA, whose update can be viewed as $x^{(t)} + \eta(x^{(t+1)} - x^{(t)})$, in which $\eta = 1$ is a constant step size. We argue that $\eta = 1$ may not be the most suitable choice and the following proposition makes this formal. We have it to theoretically analyze the update rule of ISTA and $\eta^* := \arg\min_\eta f(\eta(x^{(t+1)} - x^{(t)}) + x^{(t)}, y) + \lambda\|\eta(x^{(t+1)} - x^{(t)}) + x^{(t)}\|_1$.

**Proposition 2.** *(Requirement of overshoots) For* $\min_x f(x, y) + \lambda\|x\|_1$, *in which* $f(x, y)$ *is convex with respect to* $x$ *and* $\gamma I - \nabla_x^2 f(x) \succ 0$ *holds for all* $x$, *if the update rule in Eq. (4) is adopted, then we have* $\eta^* \geq 1$. *In addition, if* $supp(x^{(t)}) \subset supp(x^{(t+1)})$, *then we further have* $\eta^* > 1$.

See also Figure 2 for an illustration of the issue with $\eta = 1$ as concerned. Since the optimization procedure of ISTA inspires the network architecture in LISTA, the theoretical result in Proposition 2 that requires a boost in $\eta$ for superior performance also inspires us to design specific overshoot gates for LISTA. Having noticed that an essential principle we have obtained is to let $\eta \geq 1$ (or $\eta > 1$), we may expect the output of the gate function to be greater than or at least equal to 1. To achieve the goal, we can try different expressions for it, e.g.,

$$\text{the sigmoid-based function: } o_t(x^{(t)}, y|\Lambda_o^{(t)}) = 1 + a_o\sigma(W_ox^{(t)} + U_oy)\left|\sum_i y_i\right|,$$

$$\text{the inverse-proportional-based function: } o_t(x^{(t)}, y|\Lambda_o^{(t)}) = 1 + \frac{a_o}{|\tilde{x}^{(t+1)} - x^{(t)}| + \epsilon},$$

$$(18)$$

with $\sigma(\cdot)$ being the sigmoid function, $\Lambda_o^{(t)} = \{a_o, W_o, U_o\}$ and $\Lambda_o^{(t)} = \{a_o\}$ for the two types of functions respectively, and $\epsilon$ being a tiny positive constant introduced to avoid zero being divided. The principle of our overshoot gate is similar to that of some momentum-based methods, e.g., FISTA (Beck & Teboulle, 2009) and LFISTA (Moreau & Bruna, 2017). The fundamental difference between these methods and ours is that, (L)FISTA considers that the scaling factor in a momentum term should be independent of the current inputs (including the previous estimation and $y$), i.e., being time or at least input invariant, while the output of the overshoot gate is a function of both the previous estimation and $y$, hence being time-and-input-varying. The design of our overshoot gate may endow the sparse coding network higher capability to learn from its inputs. Experimental comparisons in Section 7 in the Appendix confirm the superiority of our method.

We also note that our convergence analyses in Section 3.1 generalize to $\eta > 1$ cases with a constant $\eta$, i.e., linear convergence can still be guaranteed, but the asymptotic behavior with learnable and adaptive overshoots should be further explored in future studies.

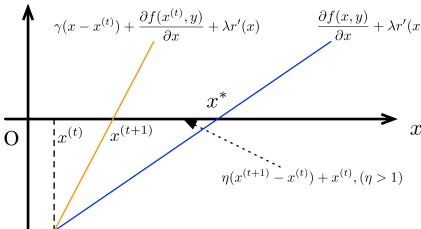

Figure 2: The derivative function (illustrated in blue) of $f(x, y) + \lambda r(x)$, in which $r(x) = \|x\|_1$, is monotonic owing to the convexity of $f(x, y)$ and $r(x)$, and its output should be consistently smaller than the derivative (illustrated in orange) of the upper bound in absolute value. Let $x^*$ be the optimal solution to the problem, then we know from the figure that the estimation with a standard ISTA update (i.e., $\eta = 1$) normally "lags behind".

## 4 EXPERIMENTS

In this section, we perform experiments to confirm our theoretical results and evaluate the performance of our gated sparse coding networks. Validations of our theoretical results are performed on synthetic data, and the performance of our method in sparse coding is tested on both synthetic and real data. We set $m = 250$, $n = 500$, and sample the elements of the dictionary matrix $A$ randomly from a standard Gaussian distribution in simulations. The position of non-zero elements of the sparse vector $x_s$ is determined by a Bernoulli sampling with a probability of 0.1 (which means approximately 90% of the elements are set to be zero). Different noise levels and condition numbers are considered in the sparse coding simulations. We randomly synthesize in-stream $x_s$ and $\varepsilon$ to obtain $y$ for training, and we let two extra sets consisting of 1000 samples each as the validation and test sets, just like in prior works[1]. (Chen et al., 2018; Liu et al., 2019; Borgerding et al., 2017).

For the proposed gated LISTA and other deep learning-based methods, we set $d = 16$ and let $\{b^{(t)}\}$ not be shared between different layers under all circumstances. The weight matrices $\{W^{(t)}, U^{(t)}\}$ are not shared either in our method and the coupled constraints $W^{(t)} = I - U^{(t)}A$, $\forall t$, are imposed. For all gates, $\nu_t$ is initialized as 1.0, and then we let the initial value of $\mu_t$ in the inverse proportional function powered gain gate be 1.0 too, since Eq. (12) and (13) indicate $0 \le \mu_t \le 2$. Other learnable parameters in our gates are uniformly initialized as 5.0 according to their suggested range of the gates. The training batch size is 64. We use Adam (Cho et al., 2014) and let $\beta_1 = 0.9$ and $\beta_2 = 0.999$. The hyper-parameters are tuned on the validation set and fixed for all our experiments in the sequel. Our training follows it of Chen et al.'s (2018). That said, the sparse coding network is trained progressively to update more layers, and we cut the learning rate for currently optimized layers when no decrease in the validation loss can be observed for 4000 iterations, with a base learning rate of 0.0005. Training on current layers stops when the validation loss does not decrease any more with the learning rate being cut to 0.00001. More details are explained in Section 8 in the appendix.

Our training objective for a network with $d$ intermediate update steps is

$$\min_{\Theta} \mathbb{E}\|x^{(d)} - x_s\|_2^2, \tag{19}$$

---

[1]The core codes of this paper could be find in github `https://github.com/wukailun/GLISTA/`.

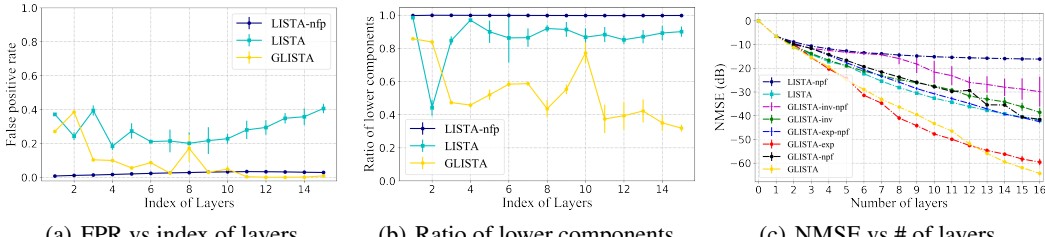

(a) FPR vs index of layers     (b) Ratio of lower components     (c) NMSE vs # of layers

Figure 3: Average results confirming our Proposition 1 and Theorem 2, in which intermediate outputs of a single network over five runs are reported in (a) and (b) while networks with varying depth are evaluated in (c), over five runs as well. It is plotted from layer indices of 1 in (a) and (b), since the two metrics (i.e., false positive rate and ratio of generated non-zero code components that require gains) do not make much sense with an initial code estimation (i.e., $\mathbf{0}$).

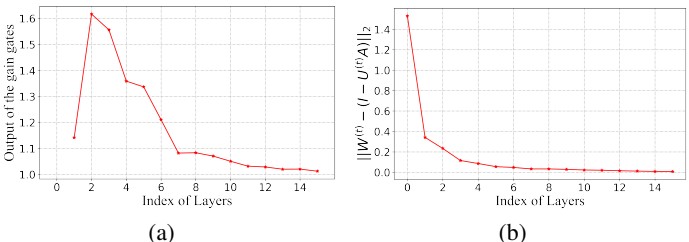

(a)         (b)

Figure 4: Average results over the whole test set confirm our Theorem 1, in which intermediate outputs of a single network is reported. It can be seen that: (a) the gate output converges to 1, and (b) LISTA with our gain gates converges as expected. It is plotted from layer index 1 in (a) since no gain is imposed on the initial code estimation (i.e., $\mathbf{0}$.)

in which $\Theta = \{W^{(t)}, U^{(t)}, b^{(t)}\}_{t=0,\ldots,d-1} \cup \Lambda^{(0)} \cup \ldots \Lambda^{(d-1)}$ is the set of all learnable parameters in the sparse coding network that generates $x^{(d)}$ given $y$. Note that in comparison with the parameter set in a standard LISTA, it also contains the parameters in gate functions. In practice, we are given a set of training samples and opt to minimize an empirical loss instead of the one in Eq. (19).

Our evaluation metric for sparse coding is the normalized MSE (NMSE) (Chen et al., 2018):

$$\text{NMSE}(x, x_s) = 10 \log_{10}(\|x - x_s\|_2^2 / \|x_s\|_2^2). \tag{20}$$

## 4.1 SIMULATION EXPERIMENTS

### 4.1.1 VALIDATION OF THEORETICAL RESULTS

**Validation of Proposition 1**: We first confirm Proposition 1. In order to ensure that LISTA fulfills the assumption about no "false positive", we introduce an auxiliary loss into the learning object as:

$$\lambda \sum_t \sum_{j \notin \text{supp}(x_s)} |x_j^{(t)}|. \tag{21}$$

We formally introduce the false positive rate (FPR) as $\text{FPR} = \frac{|\text{supp}(x^{(t)}) \cup \text{supp}(x_s)| - |\text{supp}(x_s)|}{|\text{supp}(x^{(t)})|}$ and try to approach no "false positive" (i.e., LISTA-nfp) by setting $\lambda = 5.0$ in the experiment. [2] Check Figure 3 for an illustrative comparison between different models, we see LISTA-nfp achieves almost no "false positive" in practice in Figure 3(a), but its convergence is slower as demonstrated in Figure 3(c), which is consistent with our result in **Theorem 3**. In addition, we also see in Figure 3(b) that without "false positive", the code components in LISTA estimations are almost always less than those of the ground-truth, which confirms our Proposition 1.

**Validation of Theorem 1**: We aim to calculate $\|W^{(t)}D - (I - U^{(t)}A)\|_2$ using a gated LISTA with the introduced ReLU-based piece-wise linear gain gate function [3]. To accomplish this task, we need to first evaluate the output of our gate function, which is expect to converge to 1 as shown in the theorem. We show such a trend indeed exists in Figure 4(a). Consequently, the matrix $D$ is supposed

---

[2]The FPR here is slightly different from the general false positive rate by calculating only in the obtained non-zero code components.

[3]Other functions can be adopted and the same results can be obtained.

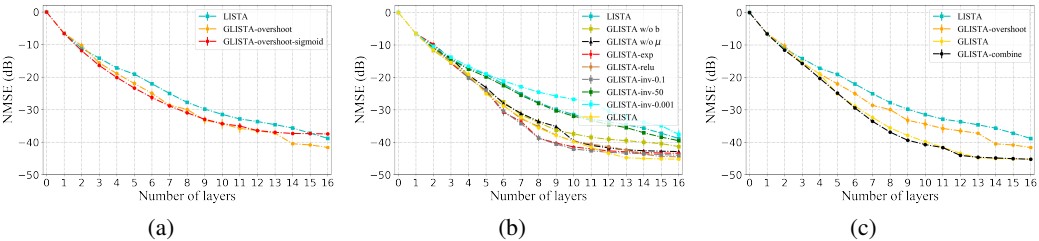

Figure 5: Comparison of different (a) overshoot gate functions, (b) gain gate functions, and (c) their combination over five runs. The experiment is performed with SNR=40dB.

to be an identity matrix in the end and we can calculate $\|W^{(t)} - (I - U^{(t)}A)\|_2$ as a surrogate. In Figure 4(b), it converges to zero in the end and the results confirm the theorem.

**Validation of Theorem 2**: We apply three kinds of gated LISTA with an alliance of gain gate functions (i.e., what has been introduced in Section 3.1.1), the exponential function, and the inverse proportional function respectively to verify our theoretical results. They were named as GLISTA (which is the abbreviation of gated LISTA), GLISTA-exp, GLISTA-inv, respectively. From Figure 3(c), we see that when the models with such gain gates has no "false positive", all of them are superior to the standard LISTA without "false positive" as well, which is consistent with the conclusion of Theorem 2. In addition, from Figure 3(a), we can also see that there actually exist "false positives" in lower layers of GLISTA, but even without the auxiliary loss term, the evaluated FPR of our GLISTA and it variants approach zero in higher layers, which is in good agreement with **Theorem 4**.

### 4.1.2 COMPARISON WITH COMPETITORS

**Empirical analyses for the gate functions:** It should be interesting to compare the performance of our gates with different expressions. We test LISTA with different overshoot gate functions introduced in Section 3.2 in Figure 5(a). Both of them are incorporated with their learnable parameters being shared among layers. It can be seen from Figure 5(a) that the accelerations in convergence and gain in final performance are obvious, just as expected. For LISTA with gain gates, one can check Figure 5(b). It can be seen that the performance degrades a lot if either the bias term or the $\mu_t$ term is removed. We also try different $f_t(\cdot)$ functions, including the ReLU-based piece-wise linear one and some possibly more nonlinear ones as mentioned in Section 3.1. We confirm that gate functions whose outputs are relatively closer to the boundary condition may perform better. Yet, it is worth noting that when the outputs of inverse proportional function reach that boundary condition and being applied uniformly to all layers, the performance degrades (see LISTA-inv-$\epsilon$ in Figure 5(b)). These results suggest an alliance of gain gate functions in practice. We further test a combination of gain gates and overshoot gates, despite the mechanism with solely gain gates is already good enough. See Figure 5(c), when overshoots are further incorporated, the convergence on lower layers becomes faster while the overall convergence is not affected much, leading to similar final performance when the model is very deep and superior performance when the model is relatively shallow.

**Compared with other state-of-the-art methods:** We consider four state-of-the-arts: LISTA with support selections (namely LISTA-C-S and LISTA-S, with and without the coupled constraint) (Chen et al., 2018), analytic LISTA with support selections (ALISTA-S) (Liu et al., 2019), and learned AMP (LAMP) (Borgerding et al., 2017) for comparison, and their official implementations are directly used. The hyper-parameters are set following the papers (Borgerding et al., 2017; Chen et al., 2018). We compare our GLISTA with these competitive methods under different levels of noises (including the signal-to-noise ratios (SNRs) being equal to 40dB, 20dB, and 10dB) and different condition numbers (including 3, 30, and 100, with SNR=40dB). See Figure 6 for comparisons between LISTA, LAMP, LISTA-S, LISTA-C-S, ALISTA-S, and our GLISTA in some of the settings. Obviously, the introduced gates facilitate LISTA significantly, and the concerned NMSE diminishes the fastest using GLISTA. See our Appendix for comparisons of final performance after multiple runs and the results in other settings (i.e., SNR: 20dB, 40dB, and condition number: 3). We know from these results that using the gain gates solely can already outperforms existing state-of-the-arts, while incorporating the overshoot gates additionally may further boost the performance, as testified.

**Applying our method to variants of LISTA:** We also try adopting the introduced gates into some variants of LISTA to verify their "generalization ability". Specifically, we incorporate the gain gates to LFISTA (Moreau & Bruna, 2017) and ALISTA (Liu et al., 2019) to obtain GFLISTA and AGLISTA, respectively. Since ALISTA is suggested to be implemented with support set selection in the original

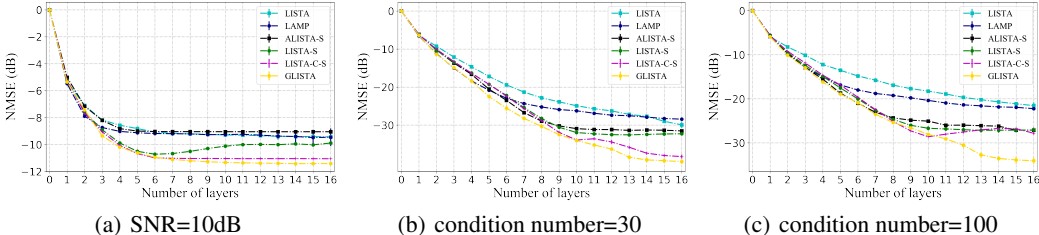

| (a) SNR=10dB | (b) condition number=30 | (c) condition number=100 |

Figure 6: Comparison of sparse coding methods in different settings over five runs. Our GLISTA consistently outperforms the competitors in almost all test cases with different numbers of layers.

paper, i.e. ALISTA-S, we also compare with it. The experiment is performed under different levels of noises (40dB, 20dB, and 10dB). As can be seen in Table 1 in which average results along with their standard deviations calculated over five runs are reported, models with our gain gates perform significantly better, which verifies that our method generalizes well.

Table 1: Comparison of LISTA and its variants (with and without gates) under different noise levels.

| SNR | LISTA | GLISTA | LFISTA | GLFISTA | ALISTA | AGLISTA | ALISTA-S |
|-----|-------|--------|--------|---------|--------|---------|----------|
| 40 | -38.72±0.09 | **-45.22±0.02** | -37.84±0.32 | **-38.30±0.10** | -37.86±0.35 | **-42.30±0.13** | -41.86±0.04 |
| 20 | -18.65±0.09 | **-23.08±0.03** | -20.90±0.02 | **-22.00±0.07** | -17.38±0.05 | **-20.13±0.03** | -20.00±0.05 |
| 10 | -9.42±0.08 | **-11.41±0.02** | -10.67±0.04 | **-11.20±0.01** | -8.39±0.04 | **-9.13±0.02** | -9.04±0.02 |

## 4.2 PHOTOMETRIC STEREO ANALYSIS

We now test on a more practical task, i.e., photometric stereo analysis, using sparse coding. For a 3D object with Lambertian surface, if there are $q$ different light conditions, a camera or some other kinds of sensors can obtain $q$ different observations, all with noises caused by shadows and specularities. The observations can be represented as a vector $o \in \mathbb{R}^q$ for estimating the norm vector $v \in \mathbb{R}^3$ at any position on the surface. It is generally formulated as $o = \rho L v + e$, in which $L \in \mathbb{R}^{q \times 3}$ represents the normalized light directions ($q$ directions), $e \in \mathbb{R}^q$ is a noise which is often sparse, $\rho \in \mathbb{R}$ represents the albedo reflectivity. Our task is to obtain $v$ from $o$ and $L$ which is also known. The estimation of $e$ can be considered as a sparse coding problem, and one can use $L^\dagger(o - e)$ to recover $v$ given the estimation. More detailed descriptions of the task can be found in Xin et al.'s paper (2016).

In the sparse coding problem, we have $Q \in \mathbb{R}^{(q-3) \times q}$ (the orthogonal complement of $L$) as the dictionary matrix (i.e., $A$ in Eq. (1)), $e$ as the sparse code to be estimated, and $Qo$ as the observation (i.e., $y$ in Eq. (1)). We mainly follow settings in Xin et al.'s work, e.g. the vectors of $L$ are randomly selected from the hemispherical surface, except that we test with $q = 15, 25, 35$, and let 40% of the elements of $e$ be non-zero. We use GLISTA here to estimate $e$ and the final result for $v$ is calculated as $L^\dagger(o - e^*)$, where $L^\dagger \in \mathbb{R}^{3 \times q}$ is the pseudo-inverse of $L$ and $e^*$ is the estimation. Our method is compared with LISTA and two traditional methods where no explicit training is introduced, i.e. the original least square (LS) and least L1, in Table 2. Our evaluation metric is the mean ($\pm$ standard deviation) error in degree and it is calculated using the bunny picture (Xin et al., 2016).

Table 2: Mean ($\pm$ standard derivation) error in degree with different number of observations over five runs.

| $q$ | LS | L1 | LISTA | GLISTA |
|-----|-----|-----|-------|--------|
| 35 | 5.37 | 1.39 | 0.0237±0.0026 | **0.00210±0.00044** |
| 25 | 5.60 | 2.03 | 0.0429±0.0068 | **0.00524±0.00024** |
| 15 | 6.09 | 4.25 | 0.371±0.046 | **0.0255±0.0054** |

## 5 CONCLUSION

In this paper, we study LISTA for solving sparse coding problems. We discover its potential weaknesses and introduce gated mechanisms to address them accordingly. In particular, we theoretically prove that LISTA with gain gates can achieve faster convergence than the standard LISTA. We also discover that LISTA (with or without gates) can obtain lower reconstruction errors under a weaker assumption of "false positive" in its code estimations. It helps us improve the convergence analyses to achieve more solid theoretical results, which have been perfectly confirmed in simulation experiments. The effectiveness of our introduced gates is verified in a variety of sparse coding experiments and the state-of-the-art performance is achieved. In the future, we aim to extend the method to convolutional neural networks to deal with more complex tasks.

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

## APPENDIX

## 6 PROOF OF THEOREMS AND PROPOSITIONS

Before we delve deeply into the proof, we first give some importance notations. We define $\mathcal{S}$ as the support of the vector $x_s$, i.e. $\mathcal{S} = \text{supp}(x_s)$, and let $|\mathcal{S}|$ denote the number of elements in the set $\mathcal{S}$. For a vector that shares the same size with $x_s$, say $z$, we denote by $z_{\mathcal{S}} \in \mathbb{R}^{|\mathcal{S}|}$ a vector that keeps the elements with indices of $z$ in $\mathcal{S}$ and removes the others. If the vectors have been introduced with subscripts already, e.g. $x_s$, we use $(x_s)_{\mathcal{S}}$ to denote vectors obtained in such a manner. For a square matrix with the same number of row and column as the size of $x_s$, say $M$, $M(\mathcal{S}, \mathcal{S})$ is its principal minor with the index set formed by removing rows and columns whose indices are not in $\mathcal{S}$. Assume a vector $x$ with no zero elements, $\text{sign}(\cdot)$ is defined as $(\text{sign}(x))_i = x_i/|x_i|$, i.e. $(\text{sign}(x))_i = 1$ when $x_i > 0$, and $(\text{sign}(x))_i = -1$ when $x_i < 0$.

### 6.1 PROOF OF PROPOSITION 1

Recall that the update rule of LISTA is $x^{(0)} = \mathbf{0}$ and

$$x^{(t+1)} = s_{b^{(t)}}(W^{(t)}x^{(t)} + U^{(t)}y), \quad t = 0, \cdots, d-1. \tag{22}$$

*Proof.* Recall the definition of $\mathcal{S}$ is $\mathcal{S} = \text{supp}(x_s)$. For the shrinking function $z = s_{b^{(t)}}(x) = \text{sign}(x)(|x| - b^{(t)})_+ = x - b^{(t)}h(z)$, where $h(z) = 1$ if $z > 0$, $h(z) = -1$ if $z < 0$, and $h(z) \in [-1, 1]$ if $z = 0$.

We use Mathematical Induction to prove $\text{supp}(x^{(t)}) \subset \mathcal{S}, \forall t = 0, 1, \ldots, d-1$. We assume $\text{supp}(x^{(t)}) \subset \mathcal{S}$. From the calculation of $x_i^{(t+1)}$, as $W^{(t)} = I - U^{(t)}A$ there is

$$
\begin{aligned}
x_i^{(t+1)} &= s_{b^{(t)}}((W^{(t)}x^{(t)} + U^{(t)}y)_i) \\
&= s_{b^{(t)}}((W^{(t)}x^{(t)} + U^{(t)}Ax_s)_i) \\
&= s_{b^{(t)}}(((I - U^{(t)}A)(x^{(t)} - x_s))_i + (x_s)_i) \\
&= ((I - U^{(t)}A)(x^{(t)} - x_s))_i + (x_s)_i - b^{(t)}h(x_i^{(t+1)}).
\end{aligned}
\tag{23}
$$

For $\forall i \notin \mathcal{S}$, $(x_s)_i = 0$. Let's assume $x_i^{(t+1)} \neq 0$, then $h(x_i^{(t+1)}) = \text{sign}(x_i^{(t+1)})$. Multiply the two sides of the Eq. (23) by $\text{sign}(x_i^{(t+1)})$, as the $b^{(t)} = \mu(A)\sup_{x_s} \|x^{(t)} - x_s\|_1$, there will be

$$
\begin{aligned}
|x_i^{(t+1)}| &= ((I - U^{(t)}A)(x^{(t)} - x_s))_i \text{sign}(x_i^{(t+1)}) - b^{(t)} \\
&= ((I - U^{(t)}A)(x^{(t)} - x_s))_i \text{sign}(x_i^{(t+1)}) - \mu(A)\sup_{x_s} \|x^{(t)} - x_s\|_1 \\
&\leq \mu(A)\|x^{(t)} - x_s\|_1 - \mu(A)\sup_{x_s} \|x^{(t)} - x_s\|_1 \leq 0,
\end{aligned}
\tag{24}
$$

the inequality holds for $\vec{a} \cdot \vec{b} \leq \|\vec{a}\|_\infty \|\vec{b}\|_1$ and $\|I - U^{(t)}A\|_\infty \leq \mu(A)$ (because of $U^{(t)} \in \mathcal{W}(A)$). From Eq. (24), we know $|x_i^{(t+1)}| \leq 0$ actually is $|x_i^{(t+1)}| = 0$ which is in conflict with $x_i^{(t+1)} \neq 0$. Therefore, the $x_i^{(t+1)} = 0$, when $i \notin \mathcal{S}$, i.e., $\text{supp}(x^{(t+1)}) \subset \mathcal{S}$. As $x^{(0)} = \mathbf{0} \subset \mathcal{S}$, the $\text{supp}(x^{(t)}) \subset \mathcal{S}, \forall t$. The no "false positive" property has been proved.

According to Eq. (23), as support set of $x_s$ and $x^{(t)}$ are the subsets of $\mathcal{S}$, there is

$$
\begin{aligned}
x_i^{(t+1)} - (x_s)_i &= ((I - U^{(t)}A)(x^{(t)} - x_s))_i - b^{(t)}h(x_i^{(t+1)}) \\
&= \sum_{j \in \mathcal{S}} (I - U^{(t)}A)_{ij}(x_j^{(t)} - (x_s)_j) - b^{(t)}h(x_i^{(t+1)}) \\
|x_i^{(t+1)} - (x_s)_i| &\leq |\sum_{j \in \mathcal{S}} (I - U^{(t)}A)_{ij}(x_j^{(t)} - (x_s)_j)| + b^{(t)}.
\end{aligned}
\tag{25}
$$

As $\mathrm{supp}(x^{(t+1)}) \subset \mathcal{S}$, accumulate all $|x_i^{(t+1)} - (x_s)_i|$ in Eq. (25) with $i \in \mathcal{S}$, there is

$$
\begin{aligned}
\|x^{(t+1)} - x_s\|_1 &\leq \sum_{i \in \mathcal{S}} \sum_{j \in \mathcal{S}} (I - U^{(t)}A)_{ij}(x_j^{(t)} - (x_s)_j) + |\mathcal{S}|b^{(t)} \\
&\leq \sum_{i \in \mathcal{S}} \sum_{j \in \mathcal{S}, i \neq j} |(I - U^{(t)}A)_{ij}||x_j^{(t)} - (x_s)_j| + |\mathcal{S}|b^{(t)} \\
&\leq (|\mathcal{S}| - 1)\mu(A)\|x^{(t)} - x_s\|_1 + |\mathcal{S}|b^{(t)}.
\end{aligned}
\tag{26}
$$

The second equation is because of $U^{(t)} \in \mathcal{W}(A)$, so that $|W_{i,:}A_{:,j}| \leq \mu(A)$ when $i \neq j$ and $|W_{i,:}A_{:,j}| = 1$ when $i = j$. Substitute $b^{(t)} = \mu(A)\sup_{x_s}\|x^{(t)} - x_s\|_1$ into Eq. (26), and take the supremum of Eq. (26). As $|\mathcal{S}| \leq s$ there is

$$
\begin{aligned}
\sup_{x_s} \|x^{(t+1)} - x_s\|_1 &\leq (s-1)\mu(A)\sup_{x_s}\|x^{(t)} - x_s\|_1 + s\mu(A)\sup_{x_s}\|x^{(t)} - x_s\|_1 \\
&\leq (2s-1)\mu(A)\sup_{x_s}\|x^{(t)} - x_s\|_1 \\
&\leq ((2s-1)\mu(A))^{t+1}\sup_{x_s}\|x^{(0)} - x_s\|_1.
\end{aligned}
\tag{27}
$$

Let $c = \log((2s-1)\mu(A))$, the $l_2$ error bound of $t$-th layer in LISTA should be calculated as

$$
\begin{aligned}
\|x^{(t)} - x_s\|_2 \leq \|x^{(t)} - x_s\|_1 &\leq \sup_{x_s}\|x^{(t)} - x_s\|_1 \\
&\leq ((2s-1)\mu(A))^t\sup_{x_s}\|x^{(0)} - x_s\|_1 \\
&= \exp(ct)\sup_{x_s}\|x^{(0)} - x_s\|_1 \\
&\leq sB\exp(ct),
\end{aligned}
\tag{28}
$$

where the last inequality is deduced since $(x_s)_i \leq B$, and $\|x_s\|_0 \leq s$. The linear convergence has been proved.

Refer to the Eq. (25), as $x_i^{(t+1)} = 0$ when $i \notin \mathcal{S}$, $|x_i^{(t+1)}| \leq |(x_s)_i|$ and $x^{(t+1)}(x_s)_i \geq 0$ hold certainly. We only concentrate on $i \in \mathcal{S}$

$$
x_i^{(t+1)} - (x_s)_i = ((I - U^{(t)}A)(x^{(t)} - x_s))_i - b^{(t)}h(x_i^{(t+1)}).
\tag{29}
$$

If $x_i^{(t+1)} = 0$, there must be $|x_i^{(t+1)}| = 0 \leq (x_s)_i$, and $x_i^{(t+1)}(x_s)_i = 0$.

If $x_i^{(t+1)} > 0$, the according to Eq. (29), $x_i^{(t+1)} - (x_s)_i = ((I - U^{(t)}A)(x^{(t)} - x_s))_i - b^{(t)} = ((I-U^{(t)}A)(x^{(t)}-x_s))_i - \mu(A)\sup_{x_s}\|x^{(t)}-x_s\|_1 \leq 0$, i.e., $0 < x_i^{(t+1)} \leq (x_s)_i$, $|x_i^{(t+1)}| \leq |(x_s)_i|$, and $x_i^{(t+1)}(x_s)_i > 0$.

If $x_i^{(t+1)} < 0$, the according to Eq. (29), $x_i^{(t+1)} - (x_s)_i = ((I - U^{(t)}A)(x^{(t)} - x_s))_i + b^{(t)} = ((I-U^{(t)}A)(x^{(t)}-x_s))_i + \mu(A)\sup_{x_s}\|x^{(t)}-x_s\|_1 \geq 0$, i.e., $0 > x_i^{(t+1)} \geq (x_s)_i$, $|x_i^{(t+1)}| \leq |(x_s)_i|$, and $x_i^{(t+1)}(x_s)_i > 0$.

In conclusion, we can obtain $|x_i^{(t+1)}| \leq |(x_s)_i|$ and $x^{(t+1)}(x_s)_i \geq 0$ for all the situations.

$\square$

## 6.2 Proof of Theorem 1

Recall that the update rule of LISTA with gain gates is $x^{(0)} = \mathbf{0}$ and

$$x^{(t+1)} = s_{b^{(t)}}(W^{(t)}(g_t(x^{(t)}, y|\Lambda_g^{(t)}) \odot x^{(t)}) + U^{(t)}y). \tag{30}$$

*Proof.* We assume that $b^{(t)}$ is a vector, i.e., $b^{(t)} \in \mathbb{R}^n$, in our proof to make it more general.

According to definition of the shrinking function $s_{b^{(t)}}(\cdot)$ and $y = Ax_s$, Eq. (10) is

$$
\begin{aligned}
x^{(t+1)} &= s_{b^{(t)}}(W^{(t)}(g_t(x^{(t)}, y|\Lambda_g^{(t)}) \odot x^{(t)}) + U^{(t)}y) \\
&= W^{(t)}g_t(x^{(t)}, y|\Lambda_g^{(t)}) \odot x^{(t)} + U^{(t)}y - b^{(t)} \odot h(x^{(t+1)}) \\
&= W^{(t)}\mathrm{diag}(g_t(x^{(t)}, y|\Lambda_g^{(t)}))x^{(t)} + U^{(t)}Ax_s - b^{(t)} \odot h(x^{(t+1)}) \\
&= W^{(t)}\mathrm{diag}(g_t(x^{(t)}, y|\Lambda_g^{(t)}))x^{(t)} + U^{(t)}Ax_s - b^{(t)} \odot h(x^{(t+1)}).
\end{aligned}
\tag{31}
$$

Concentrate on the situation of $t \to \infty$. Define $g_\kappa(x_s) = g_t(x_s, y|\Lambda_g^{(t)})$ when $t \to \infty$. In the main body of Theorem 1, $\forall x_s$ satisfying $\|x_s\|_0 \leq s$ is the fixed point of Eq. (10) when $t \to \infty$. Eq. (31) is

$$x_s = W^{(t)}\mathrm{diag}(g_\kappa(x_s))x_s + U^{(t)}Ax_s - b^{(t)} \odot h(x_s). \tag{32}$$

The equation group of the indices in $\mathcal{S}$ in Eq. (32) is

$$
\begin{aligned}
(x_s)_\mathcal{S} &= (((W^{(t)}\mathrm{diag}(g_\kappa(x_s)) + U^{(t)}A)x_s)_\mathcal{S} - b_\mathcal{S}^{(t)} \odot h((x_s)_\mathcal{S}) \\
&= (W^{(t)}(\mathcal{S}, \mathcal{S})\mathrm{diag}(g_\kappa((x_s)_\mathcal{S})) + (U^{(t)}A)(\mathcal{S}, \mathcal{S}))(x_s)_\mathcal{S} - b_\mathcal{S}^{(t)} \odot h((x_s)_\mathcal{S}).
\end{aligned}
\tag{33}
$$

Let $(x_s)_\mathcal{S} \to 0$ but $(x_s)_\mathcal{S} \neq 0$ so that $h((x_s)_\mathcal{S}) = \mathrm{sign}((x_s)_\mathcal{S})$. As $W^{(t)}$, $U^{(t)}$, $A$ and $g_\kappa(x_s) = g_t(x_s, y|\Lambda_g^{(t)})$ are bounded, the right hand side of Eq. (33) is also tend to 0, which is

$$b_\mathcal{S}^{(t)} \to 0, \quad \text{as} \quad t \to \infty. \tag{34}$$

As the $\mathcal{S}$ can be selected arbitrarily as long as $|\mathcal{S}| \leq s$, $b^{(t)}$ also satisfies

$$b^{(t)} \to 0, \quad \text{as} \quad t \to \infty. \tag{35}$$

Substitute the $b_\mathcal{S}^{(t)}$ of Eq. (34) into Eq. (33), $(x_s)_\mathcal{S}$ is

$$(x_s)_\mathcal{S} = (W^{(t)}(\mathcal{S}, \mathcal{S})\mathrm{diag}(g_\kappa((x_s)_\mathcal{S}))_\mathcal{S} + (U^{(t)}A)(\mathcal{S}, \mathcal{S}))(x_s)_\mathcal{S}, \tag{36}$$

where the $W^{(t)}(\mathcal{S}, \mathcal{S})$ is defined at start of this section. Eq. (36) is

$$
\begin{aligned}
(I - U^{(t)}A)(\mathcal{S}, \mathcal{S})(x_s)_\mathcal{S} &= W^{(t)}(\mathcal{S}, \mathcal{S})\mathrm{diag}(g_\kappa((x_s)_\mathcal{S}))(x_s)_\mathcal{S}, \\
(I - U^{(t)}A)(\mathcal{S}, \mathcal{S})(x_s)_\mathcal{S} &= W^{(t)}(\mathcal{S}, \mathcal{S})\mathrm{diag}((x_s)_\mathcal{S})g_\kappa((x_s)_\mathcal{S}), \\
g_\kappa((x_s)_\mathcal{S}) &= \mathrm{diag}(((x_s)_\mathcal{S})^{-1})(W^{(t)}(\mathcal{S}, \mathcal{S}))^{-1}(I - U^{(t)}A)(\mathcal{S}, \mathcal{S})(x_s)_\mathcal{S},
\end{aligned}
\tag{37}
$$

$$\mathrm{diag}(((x_s)_\mathcal{S})^{-1})M(x_s)_\mathcal{S} = g_\kappa((x_s)_\mathcal{S}),$$

where $M = (W^{(t)}(\mathcal{S}, \mathcal{S}))^{-1}(I - U^{(t)}A)(\mathcal{S}, \mathcal{S})$. The $i$-th row and $j$-th column element in $M$ is denoted as $m_{ij}$. From Eq. (37), $(g_\kappa(x_s))_\mathcal{S}$ is

$$
(g_\kappa(x_s))_\mathcal{S} =
\begin{bmatrix}
((x_s)_\mathcal{S})_1^{-1} & & & \\
& ((x_s)_\mathcal{S})_2^{-1} & & \\
& & \ldots & \\
& & & ((x_s)_\mathcal{S})_{|\mathcal{S}|}^{-1}
\end{bmatrix}
\begin{bmatrix}
m_{11} & m_{12} & \ldots & m_{1|\mathcal{S}|} \\
m_{21} & m_{22} & \ldots & m_{2|\mathcal{S}|} \\
\ldots & \ldots & \ldots & \ldots \\
m_{|\mathcal{S}|1} & m_{|\mathcal{S}|2} & \ldots & m_{|\mathcal{S}||\mathcal{S}|}
\end{bmatrix}
$$

$$
\begin{bmatrix}
((x_s)_\mathcal{S})_1 \\
((x_s)_\mathcal{S})_2 \\
\ldots \\
((x_s)_\mathcal{S})_{|\mathcal{S}|}
\end{bmatrix}
$$

$$
=
\begin{bmatrix}
\frac{\sum_{i=1}^{|\mathcal{S}|} m_{1i}((x_s)_\mathcal{S})_i}{((x_s)_\mathcal{S})_1} \\
\frac{\sum_{i=1}^{|\mathcal{S}|} m_{2i}((x_s)_\mathcal{S})_i}{((x_s)_\mathcal{S})_2} \\
\ldots \\
\frac{\sum_{i=1}^{|\mathcal{S}|} m_{|\mathcal{S}|i}((x_s)_\mathcal{S})_i}{((x_s)_\mathcal{S})_{|\mathcal{S}|}}
\end{bmatrix},
$$

where this equation should hold for all $x_s$ in Assumption 1. Assume $(x_s)_\mathcal{S} \to 0$, for $g_\kappa(x_s)$ is bounded, we can conclude that $m_{ij} = 0$, if $i \neq j$. From Eq. (38), the final form of $g_\kappa((x_s)_\mathcal{S})$ is formulated as

$$g_\kappa((x_s)_\mathcal{S}) = \begin{bmatrix} m_{11} \\ m_{22} \\ ... \\ m_{|\mathcal{S}||\mathcal{S}|} \end{bmatrix}. \tag{38}$$

From Eq. (38), we can conclude that $g_\kappa(x_s)_i$ is a constant if $i \in \mathcal{S}$, as the $\mathcal{S}$ could be arbitrary subset of $\{1, \cdots, n\}$ as long as $|\mathcal{S}| \leq s$. We could deduce that $g_\kappa(x_s)_i$ is constant $\forall i \in \{1, \ldots, n\}$ and $g_\kappa(x_s)$ must be constant vector, i.e.

$$\mathrm{diag}(g_\kappa(x_s)) = D, \tag{39}$$

where $D$ is an $n \times n$ constant diagonal matrix. The first part of conclusion of Theorem 1 has been proved.

Substitute $b^{(t)}$ in Eq. (34) and $\mathrm{diag}(g_\kappa(x_s))$ in Eq. (39) into Eq. (32), Eq. (32) is.

$$\begin{aligned} x_s &= (W^{(t)}D + U^{(t)}A)x_s, \\ x_s &= Zx_s, \end{aligned} \tag{40}$$

where $Z = W^{(t)}D + U^{(t)}A = [Z_1, Z_2, \ldots, Z_n]$ and the $Z_i$ is the $i$-th column of $Z$.

Give a $x_s$ satisfying only the $i$-th element of $x_s$ is non-zero and all the other elements are equal to zero, i.e., $x_s = [0, 0, \ldots, \omega, \ldots, 0]^T = \omega e_i$, in which $e_i$ is basis vector with only the $i$-th element being 1 and $\omega \neq 0$. Substitute the $x_s = \omega e_i$ into Eq. (40), there is

$$\begin{aligned} x_s &= Zx_s, \\ \omega e_i &= [Z_1, Z_2, \ldots, Z_n][0, 0, \ldots, \omega, \ldots, 0]^T, \\ \omega e_i &= \omega Z_i, \\ \omega(e_i - Z_i) &= \mathbf{0}. \end{aligned} \tag{41}$$

As the Eq. (41) should hold for $\forall \omega \neq 0$, we can deduce that $Z_i = e_i$. As the $i$ is selected arbitrarily, $Z = W^{(t)}D + U^{(t)}A = [Z_1, Z_2, \ldots, Z_n] = [e_1, e_2, \ldots, e_n] = I$. Thus we have completed the proof and get

$$W^{(t)}D = (I - U^{(t)}A) \quad \text{as} \quad t \to \infty. \tag{42}$$

$\square$

### 6.3 PROOF OF THEOREM 2

Recall that the update rule of LISTA with gain gates is $x^{(0)} = \mathbf{0}$ and

$$x^{(t+1)} = s_{b^{(t)}}(W^{(t)}(g_t(x^{(t)}, y|\Lambda_g^{(t)}) \odot x^{(t)}) + U^{(t)}y). \tag{43}$$

*Proof.* We simplify the $g_t(x^{(t)}, y|\Lambda_g^{(t)})$ as $g_t(x^{(t)})$, and $\kappa_t(x^{(t)}, y|\Lambda_g^{(t)})$ as $\kappa_t(x^{(t)})$. According to the definition of gain gate in Eq. (43), we have

$$\begin{aligned} x^{(t+1)} &= s_{b^{(t)}}(W^{(t)}(x^{(t)} \odot g_t(x^{(t)}) + U^{(t)}y) \\ &= s_{b^{(t)}}(W^{(t)}(x^{(t)} \odot g_t(x^{(t)}) + U^{(t)}Ax_s) \\ &= s_{b^{(t)}}((I - U^{(t)}A)(x^{(t)} \odot g(x^{(t)}) - x_s) + x_s) \\ &= (I - U^{(t)}A)(x^{(t)} \odot g(x^{(t)}) - x_s) + x_s - b^{(t)}h(x^{(t+1)}). \end{aligned} \tag{44}$$

Simplify the $x^{(t)} \odot g(x^{(t)}) - x_s$ as $\Delta_g x^{(t)}$. For the $i$-th equation in Eq. (44), and $i \notin \mathcal{S}$, give the value of $b^{(t)} = \mu(A)\sup_{x_s} \|x^{(t)} \odot g(x^{(t)}) - x_s\|_1$, there is

$$\begin{aligned} x_i^{(t+1)} &= ((I - U^{(t)}A)(\Delta_g x^{(t)}))_i - b^{(t)}h(x_i^{(t+1)}) \\ &= ((I - U^{(t)}A)(\Delta_g x^{(t)}))_i - \mu(A)\sup_{x_s} \|\Delta_g x^{(t)}\|_1 h(x_i^{(t+1)}). \end{aligned} \tag{45}$$

With almost the same proof process in Proposition 1, we could deduce that

$$\text{supp}(x^{(t+1)}) \subset x_s, \tag{46}$$

which is the no "false poistive" property.

Recall the Eq. (44) and substitute the $1 + \kappa_{t+1}(x^{(t+1)}) = g_{t+1}(x^{(t+1)})$ into it:

$$x^{(t+1)} = ((I - U^{(t)}A)(\Delta_g x^{(t)})) - b^{(t)}h(x^{(t+1)}) + x_s,$$

$$x^{(t+1)}(1 + \kappa_{t+1}(x^{(t+1)})) = (I - U^{(t)}A)(\Delta_g x^{(t)}) - b^{(t)}h(x^{(t+1)})$$
$$+ x_s + x^{(t+1)} \odot \kappa_t(x^{(t+1)}), \tag{47}$$

$$\Delta_g x^{(t+1)} = (I - U^{(t)}A)(\Delta_g x^{(t)}) - b^{(t)}h(x^{(t+1)}) + x^{(t+1)} \odot \kappa_t(x^{(t+1)}).$$

We shall calculate the non-zero $|\Delta_g x_i^{(t+1)}|$ with the index $i$. The $i$ could be seperated to two parts. One is $i \in \mathcal{S}$ but $i \notin \text{supp}(x^{(t+1)})$, another one part is $i \in \mathcal{S}$ and $i \in \text{supp}(x^{(t+1)})$. Two kinds of $i$ are discussed respectively.

For $i \in \mathcal{S}$ but $i \notin \text{supp}(x^{(t+1)})$, there must be $x_i^{(t+1)} = 0$ and $-1 \le h(x_i^{(t+1)}) \le 1$. Select the $i$-th equation in Eq. (47), there is

$$|\Delta_g x_i^{(t+1)}| = |((I - U^{(t)}A)(\Delta_g x^{(t)}))_i - b^{(t)}h(x_i^{(t+1)})|$$
$$\le \mu(A) \sum_{j \in \mathcal{S}, j \ne i} |\Delta_g x_j^{(t)}| + |b^{(t)}|. \tag{48}$$

For $i \in \mathcal{S}$ and $i \in \text{supp}(x^{(t+1)})$, there must be $x^{(t+1)} \ne 0$ and $h(x^{(t+1)}) = \text{sign}(x^{(t+1)})$. Select the $i$-th equation in Eq. (47), there is

$$|\Delta_g x_i^{(t+1)}| = ((I - U^{(t)}A)(\Delta_g x^{(t)}))_i - b^{(t)}h(x_i^{(t+1)}) + x_i^{(t+1)}\kappa_{t+1}(x_i^{(t+1)})$$
$$\le \mu(A) \sum_{j \in \mathcal{S}, j \ne i} |\Delta_g x_j^{(t)}| - b^{(t)}\text{sign}(x_i^{(t+1)})$$
$$+ \text{sign}(x_i^{(t+1)})(|x_i^{(t+1)}|\kappa_{t+1}(x_i^{(t+1)})) \tag{49}$$
$$= \mu(A) \sum_{j \in \mathcal{S}, j \ne i} |\Delta_g x_j^{(t)}| + (|x_i^{(t+1)}|\kappa_{t+1}(x_i^{(t+1)}) - b^{(t)})\text{sign}(x_i^{(t+1)}).$$

According to the condition in Eq. (12) and (13), the $0 < \kappa_t(x) \odot |x| < 2b^{(t-1)}$. Then, $|\kappa_t(x) \odot |x| - b^{(t-1)}| < b^{(t-1)}$, there must $\exists \eta < 1$, so that $|\kappa_t(x) \odot |x| - b^{(t-1)}| \le \eta b^{(t-1)} < b^{(t-1)}$. Substituting it to Eq. (49), there is

$$|\Delta_g x_i^{(t+1)}| \le \mu(A) \sum_{j \in \mathcal{S}, j \ne i} |\Delta_g x_i^{(t)}| + (|x_i^{(t+1)}|\kappa_{t+1}(x_i^{(t+1)}) - b^{(t)})\text{sign}(x_i^{(t+1)})$$
$$\le \mu(A) \sum_{j \in \mathcal{S}, j \ne i} |\Delta_g x_i^{(t)}| + ||x_i^{(t+1)}|\kappa_{t+1}(x_i^{(t+1)}) - b^{(t)}| \tag{50}$$
$$\le \mu(A) \sum_{j \in \mathcal{S}, j \ne i} |\Delta_g x_i^{(t)}| + \eta b^{(t)}.$$

Accumulate all the $|\Delta_g x_i^{(t+1)}|$ with all $i \in \mathcal{S}$, and define $s^{(t)} = |\text{supp}(x^{(t)})|$ as the number of non-zeros elements in $x^{(t)}$ there is

$$\|\Delta_g x^{(t+1)}\|_1 \le \sum_{i \in \mathcal{S}} \mu(A) \sum_{j \in \mathcal{S}, j=i} |\Delta_g x_i^{(t)}| + (s^{(t+1)}\eta + (|\mathcal{S}| - s^{(t+1)}))b^{(t)}$$
$$\le \sum_{i \in \mathcal{S}} \mu(A) \sum_{j \in \mathcal{S}, j=i} |\Delta_g x_i^{(t)}| + (s^{(t+1)}\eta + (|\mathcal{S}| - s^{(t+1)}))b^{(t)} \tag{51}$$
$$\le (|\mathcal{S}| - 1)\mu(A)\|\Delta_g x^{(t)}\|_1$$
$$+ (s^{(t+1)}\eta + (|\mathcal{S}| - s^{(t+1)}))\mu(A) \sup_{x_s} \|\Delta_g x^{(t)}\|_1.$$

Take the supremum of Eq. (51), let $s_*^{(t)}$ denote the infimum of $s^{(t)}$ with all of the $x_s \in \mathcal{X}(B, s, 0)$, and $c_t = \log((2s - 1 - s_*^{(t)}(1 - \eta))\mu(A))$, there is

$$
\begin{aligned}
\sup_{x_s} \|\Delta_g x^{(t+1)}\|_1 &\leq (s - 1)\mu(A) \sup_{x_s} \|\Delta_g x^{(t)}\|_1 \\
&\quad + (s_*^{(t+1)}\eta + (s - s_*^{(t+1)}))\mu(A) \sup_{x_s} \|\Delta_g x^{(t)}\|_1 \\
&\leq (2s - 1 + s_*^{(t+1)}(1 - \eta)) \sup_{x_s} \|\Delta_g x^{(t)}\|_1 \\
&\leq \exp(c_{t+1}) \sup_{x_s} \|\Delta_g x^{(t)}\|_1 \\
&\leq \exp(\sum_{i=1}^{t+1} c_i) \sup_{x_s} \|\Delta_g x^{(0)}\|_1 \leq \exp(\sum_{i=1}^{t+1} c_i) sB.
\end{aligned}
\tag{52}
$$

Eq. (52) gives the upper bound of $\sup_{x_s} \|\Delta_g x^{(t+1)}\|_1$, next we shall deduce the relationship between $\|x^{(t)} - x_s\|_1$ and it. For the last layer ($t$-th layer), from Eq. (44), we have

$$
\begin{aligned}
x_i^{(t+1)} &= ((I - U^{(t)}A)(\Delta_g x^{(t)}))_i - \mu(A) \sup_{x_s} \|\Delta_g x^{(t)}\|_1 h(x_i^{(t+1)}) + (x_s)_i, \\
x_i^{(t+1)} - (x_s)_i &= ((I - U^{(t)}A)(\Delta_g x^{(t)}))_i - \mu(A)h(x_i^{(t+1)}) \sup_{x_s} \|\Delta_g x^{(t)}\|_1, \\
|x_i^{(t+1)} - (x_s)_i| &\leq |((I - U^{(t)}A)(\Delta_g x^{(t)}))_i| + \mu(A) \sup_{x_s} \|\Delta_g x^{(t)}\|_1.
\end{aligned}
\tag{53}
$$

Using almost the same process in Eq. (26) and Eq. (27), we could deduce Eq. (53) that

$$
\begin{aligned}
\|x^{(t+1)} - x_s\|_1 &\leq (2s - 1)\mu(A) \sup_{x_s} \|\Delta_g x^{(t)}\|_1, \\
\|x^{(t)} - x_s\|_1 &\leq (2s - 1)\mu(A) \sup_{x_s} \|\Delta_g x^{(t-1)}\|_1 \\
&\leq (2s - 1)\mu(A) \exp(\sum_{i=1}^{t-1} c_i) sB \\
&= \exp(\sum_{i=1}^{t-1} c_i + c) sB,
\end{aligned}
\tag{54}
$$

where the third inequality sign holds because of Eq. (52) and the last equation holds because of $c = \log((2s - 1)\mu(A))$.

As at least $c_t \leq c$ satisfies, there will be $\sup_{x_s} \|x^{(t)} - x_s\|_1 \leq \exp(ct)sB$. Set $t_0 = \lceil \log(\frac{sB}{\|x_s\|_1}) / \log(\frac{1}{(2s-1)\mu(A)}) \rceil$. When $i > t_0$, $\sup_{x_s} \|x^{(i)} - x_s\|_1 \leq \exp(ci)sB < \|x_s\|_1$, $\|x^{(i)} - x_s\|_1 < \|x_s\|_1$, then the $s_*^{(i)} = |\text{supp}(x^{(i)})| > 0$, $c_i = \log(2s - 1 + s_*^{(i)}(1 - \eta))\mu(A) < \log((2s - 1)\mu(A)) = c$.

In conclusion, from Eq. (54), there is

$$
\|x^{(t)} - x_s\|_2 \leq \|x^{(t)} - x_s\|_1 \leq \exp(\sum_{i=1}^{t-1} c_i + c) sB,
\tag{55}
$$

where $c = \log((2s - 1)\mu(A))$, $c_i = c$ when $i \leq t_0$, and $c_i < c$ when $i > t_0$. □

### 6.4 PROOF OF THEOREM 3

*Proof.* For the $t$-th layer of the LISTA, according to the Eq. (23), we have

$$
x_i^{(t+1)} = ((I - U^{(t)}A)(x^{(t)} - x_s))_i + (x_s)_i - b^{(t)}h(x_i^{(t+1)}).
\tag{56}
$$

As we have remove the no false positive assumption, $\text{supp}(x_i^{(t)}) \nsubseteq \mathcal{S}$. Define $\mathcal{S}^{(t)}$ as $\forall i \in \mathcal{S}^{(t)}$ satisfies $i \in \text{supp}(x^{(t)})$ but $i \notin \mathcal{S}$. In order to calculate all of non-zero $x_i^{(t+1)} - (x_s)_i$ with index $i$, we divide the $i$ into two kinds, $i \in \mathcal{S}$ and $i \in \mathcal{S}^{(t+1)}$.

If $i \in \mathcal{S}$, from Eq. (56), we can deduce the same formulation as Eq. (25):

$$
\begin{aligned}
|x_i^{(t+1)} - (x_s)_i| &\leq \sum_{j \in \text{supp}(x^{(t)})} (I - U^{(t)}A)_{ij}(x_j^{(t)} - (x_s)_j) + b^{(t)} \\
&\leq \sum_{j \in \text{supp}(x^{(t)})} |(I - U^{(t)}A)_{ij}(x_j^{(t)} - (x_s)_j)| + b^{(t)}.
\end{aligned}
\tag{57}
$$

If $i \in \mathcal{S}^{(t+1)}$, then the $(x_s)_i = 0$ but $x_i^{(t+1)}$, then $h(x_i^{(t+1)}) = \text{sign}(x_i^{(t+1)})$. From Eq. (56), there is

$$
\begin{aligned}
x_i^{(t+1)} &= ((I - U^{(t)}A)(x^{(t)} - x_s))_i - b^{(t)}h(x_i^{(t+1)}) \\
&= ((I - U^{(t)}A)(x^{(t)} - x_s))_i - b^{(t)}\text{sign}(x_i^{(t+1)}).
\end{aligned}
\tag{58}
$$

Multiply $\text{sign}(x_i^{(t+1)})$ on Eq. (58), we have

$$
\begin{aligned}
|x_i^{(t+1)}| &= ((I - U^{(t)}A)(x^{(t)} - x_s))_i\text{sign}(x_i^{(t+1)}) - b^{(t)}, \\
|x_i^{(t+1)}| + b^{(t)} &= ((I - U^{(t)}A)(x^{(t)} - x_s))_i\text{sign}(x_i^{(t+1)}), \\
(|x_i^{(t+1)}| + b^{(t)})\text{sign}(x_i^{(t+1)}) &= ((I - U^{(t)}A)(x^{(t)} - x_s))_i,
\end{aligned}
\tag{59}
$$

which means $((I - U^{(t)}A)(x^{(t)} - x_s))_i$ have the same sign with $\text{sign}(x_i^{(t+1)})$ because $|x_i^{(t+1)}| > 0$ and $b^{(t)} > 0$, i.e.,

$$
((I - U^{(t)}A)(x^{(t)} - x_s))_i = |((I - U^{(t)}A)(x^{(t)} - x_s))_i|\text{sign}(x_i^{(t+1)}).
\tag{60}
$$

From the Eq. (58), substitute Eq. (60) into Eq. (58), there is

$$
\begin{aligned}
x_i^{(t+1)} - (x_s)_i &= x_i^{(t+1)} \\
&= \text{sign}(x_i^{(t+1)})(|\sum_{j \in \text{supp}(x^{(t)})} (I - U^{(t)}A)_{ij}(x_j^{(t)} - (x_s)_j)| \\
&\quad - b^{(t)}), \\
|x_i^{(t+1)} - (x_s)_i| = |x_i^{(t+1)}| = x_i^{(t+1)}\text{sign}(x_i^{(t+1)}) \\
&= |\sum_{j \in \text{supp}(x^{(t)})} (I - U^{(t)}A)_{ij}(x_j^{(t)} - (x_s)_j)| - b^{(t)} \\
&\leq \sum_{j \in \text{supp}(x^{(t)})} |(I - U^{(t)}A)_{ij}(x_j^{(t)} - (x_s)_j)| - b^{(t)}
\end{aligned}
\tag{61}
$$

Accumulate all the $|x_i^{(t+1)} - (x_s)_i|$ with $i \in \text{supp}(x^{(t+1)}) \cup \text{supp}(x_s) = \mathcal{S}^{(t+1)} + \mathcal{S}$, there is

$$
\begin{aligned}
\|x^{(t+1)} - x_s\|_1 &\leq \sum_{i \in \mathcal{S}^{(t+1)}+\mathcal{S}} \sum_{j \in \text{supp}(x^{(t)})} |(I - U^{(t)}A)_{ij}(x_j^{(t)} - (x_s)_j)| \\
&\quad + (|\mathcal{S}| - |\mathcal{S}^{(t+1)}|)|b^{(t)}|, \\
&\leq (|\mathcal{S}^{(t+1)}| + |\mathcal{S}|)\mu(A)\|x^{(t)} - x_s\|_1 + (|\mathcal{S}| - |\mathcal{S}^{(t+1)}|)b^{(t)}.
\end{aligned}
\tag{62}
$$

Substitute the $b^{(t)} = \omega_{t+1}(k_{t+1}|\Theta)\mu(A)\sup_{x_s}\|x^{(t)} - x_s\|_1$ into Eq. (62), and take its supremum of right part:

$$
\begin{aligned}
\|x^{(t+1)} - x_s\|_1 &\leq (|\mathcal{S}^{(t+1)}| + |\mathcal{S}|)\mu(A)\sup_{x_s}\|x^{(t)} - x_s\|_1 \\
&\quad + (|\mathcal{S}| - |\mathcal{S}^{(t+1)}|)\omega_{t+1}(k_{t+1}|\Theta)\mu(A)\sup_{x_s}\|x^{(t)} - x_s\|_1 \\
&\leq (|\mathcal{S}^{(t+1)}| + |\mathcal{S}| + (|\mathcal{S}| - |\mathcal{S}^{(t+1)}|)\omega_{t+1}(k_{t+1}|\Theta)) \\
&\quad \mu(A)\sup_{x_s}\|x^{(t)} - x_s\|_1.
\end{aligned}
\tag{63}
$$

Take the supremum of left part of (63), there is

$$\sup_{x_s} \|x^{(t+1)} - x_s\|_1 \le \exp(c^*_{t+1}) \sup_{x_s} \|x^{(t)} - x_s\|_1, \tag{64}$$

where $c^*_{t+1} = \sup_{x_s} \log((|\mathcal{S}^{(t+1)}| + |\mathcal{S}| + (|\mathcal{S}| - |\mathcal{S}^{(t+1)}|)\omega_{t+1}(k_{t+1}|\Theta))\mu(A))$. According to the definition of $\omega_t(\cdot)$, $b^{(t)} = \omega_{t+1}(k_{t+1}|\Theta)\mu(A)\sup_{x_s}\|x^{(t)} - x_s\|_1$, so that the number of false positive is less or equal than $k_{t+1}$, i.e. $|\mathcal{S}^{(t+1)}| \le k_{t+1}$. According to the previous proof, when $b^{(t)} = \mu(A)\sup_{x_s}\|x^{(t)} - x_s\|_1$, the number of false positive satisfies $k_{t+1} = 0$. That means that $\omega(0|\Theta) \le 1$ and $\omega(k|\Theta) \le 1$ when $k > 0$ [4]. The $c^*_{t+1}$ should be

$$\begin{aligned} c^*_{t+1} &= \sup_{x_s} \log((|\mathcal{S}^{(t+1)}| + |\mathcal{S}| + (|\mathcal{S}| - |\mathcal{S}^{(t+1)}|)\omega_{t+1}(k_{t+1}|\Theta))\mu(A)) \\ &= \log((s + k_{t+1} + (s - k_{t+1})\omega_{t+1}(k_{t+1}|\Theta))\mu(A)). \end{aligned} \tag{65}$$

As assumption of $\omega_{t+1}(\cdot|\Theta)$, $\exists k_0^{t+1}$, s.t. $0 < k_0^{t+1} < s$, and $\omega_{t+1}(k_0^{t+1}|\Theta) < 1 - 1/(s - k_0^{t+1})$. Select the value of $b^{(t)}$ so that $k_{t+1} = k_0^{t+1}$, we substitute it to Eq. (65), there will be

$$\begin{aligned} c^*_{t+1} &= \log((s + k_{t+1} + (s - k_{t+1})\omega_{t+1}(k_{t+1}|\Theta))\mu(A)) \\ &< \log((s + k_{t+1} + (s - k_{t+1})(1 - \frac{1}{s - k_{t+1}}))\mu(A)) \\ &= \log((2s - 1)\mu(A)). \end{aligned} \tag{66}$$

Recall the Eq. (64), we have

$$\begin{aligned} \sup_{x_s} \|x^{(t+1)} - x_s\|_1 &\le \exp(c^*_{t+1}) \sup_{x_s} \|x^{(t)} - x_s\|_1 \\ &\le \exp(\sum_{i=1}^{t+1} c^*_i) \sup_{x_s} \|x^{(0)} - x_s\|_1 \\ &\le \exp(\sum_{i=1}^{t+1} c^*_i) sB. \end{aligned} \tag{67}$$

The $l_2$ error bound of the $t$-th layer of LISTA is

$$\|x^{(t)} - x_s\|_2 \le \|x^{(t)} - x_s\|_1 \le \sup_{x_s} \|x^{(t)} - x_s\|_1 \le sB \exp(\sum_{i=1}^{t} c^*_i), \tag{68}$$

where $c^*_i < \log((2s - 1)\mu(A))$.

$\square$

## 6.5 PROOF OF THEOREM 4

*Proof.* For the $t$-th layer given in Eq. (10), according to Eq. (47),

$$x^{(t+1)} = ((I - U^{(t)}A)(\Delta_g x^{(t)})) - b^{(t)} h(x^{(t+1)}) + x_s, \tag{69}$$

and

$$\Delta_g x^{(t+1)} = (I - U^{(t)}A)(\Delta_g x^{(t)}) - b^{(t)} h(x^{(t+1)}) + x^{(t+1)} \odot \kappa_t(x^{(t+1)}). \tag{70}$$

As the no false positive is not fit for $x^{(t)}$, $x^{(t)} \nsubseteq \mathcal{S}$. We still define $\mathcal{S}^{(t)}$ as $\forall i \in \mathcal{S}^{(t)}$ satisfies $i \in \text{supp}(x_i^{(t)})$ but $i \notin \mathcal{S}$ and define $\mathbb{S}^{(t)}$ as $\forall i \in \mathbb{S}^{(t)}$ satisfies $i \in \mathcal{S}$ and $i \in \text{supp}(x^{(t)})$. In order to calculate the non-zero $\Delta_g x_i^{(t+1)}$, we divide the $i$ into three situations: $i \in \mathbb{S}^{(t+1)}$, $i \notin \mathbb{S}^{(t+1)}$ but $i \in \mathcal{S}$, and $i \in \mathcal{S}^{(t+1)}$.

---

[4] According to the definition of $\omega_t$, $\omega_t$ must be a monotonic decreasing function and $\omega_t(k|\Theta) < 1$ when $k > 0$.

For $i \in \mathbb{S}^{(t+1)}$, there must be $x_i^{(t+1)} \neq 0$ , and $(x_s)_i \neq 0$. Substitute the form of $\kappa_t$ into $i$-th equation of Eq. (70):

$$
\begin{aligned}
\Delta_g x_i^{(t+1)} &= ((I - U^{(t)}A)(\Delta_g x^{(t)}))_i - b^{(t)}\text{sign}(x^{(t+1)}) + \mu_{t+1}b^{(t)}\text{sign}(x_i^{(t+1)}) \\
&= ((I - U^{(t)}A)(\Delta_g x^{(t)}))_i - (1 - \mu_{t+1})b^{(t)}\text{sign}(x^{(t+1)}), \\
|\Delta_g x_i^{(t+1)}| &\leq |((I - U^{(t)}A)(\Delta_g x^{(t)}))_i| + (1 - \mu_{t+1})b^{(t)},
\end{aligned}
\tag{71}
$$

we have assume $\mu_t \leq 1$

For $i \notin \mathbb{S}^{(t+1)}$ but $i \in \mathcal{S}$, $x_i^{(t+1)} = 0$, and $(x_s)_i \neq 0$. The $i$-th equation of Eq. (70) is

$$
\begin{aligned}
\Delta_g x_i^{(t+1)} &= ((I - U^{(t)}A)(\Delta_g x^{(t)}))_i - b^{(t)}\text{sign}(x_i^{(t+1)}), \\
|\Delta_g x_i^{(t+1)}| &\leq |((I - U^{(t)}A)(\Delta_g x^{(t)}))_i| + b^{(t)}.
\end{aligned}
\tag{72}
$$

For $i \in \mathcal{S}^{(t+1)}$, $(x_s)_i = 0$.

$$
\Delta_g x_i^{(t+1)} = x_i^{(t+1)} g_t(x_i^{(t+1)}) = ((I - U^{(t)}A)(\Delta_g x^{(t)}))_i - (1 - \mu_{t+1})b^{(t)}\text{sign}(x^{(t+1)}), \tag{73}
$$

As $g_t(x_i^{(t+1)}) \geq 1$, the sign of $x_i^{(t+1)} g_t(x_i^{(t+1)})$ is the same as that of $x_i^{(t+1)}$. Multiply $\text{sign}(x_i^{(t+1)})$ on Eq. (73), there is

$$
\begin{aligned}
|x_i^{(t+1)} g_t(x_i^{(t+1)})| &= ((I - U^{(t)}A)(\Delta_g x^{(t)}))_i\text{sign}(x^{(t+1)}) - (1 - \mu_{t+1})b^{(t)}, \\
|x_i^{(t+1)} g_t(x_i^{(t+1)})| + (1 - \mu_{t+1})b^{(t)} &= ((I - U^{(t)}A)(\Delta_g x^{(t)}))_i\text{sign}(x^{(t+1)}),
\end{aligned}
\tag{74}
$$

which means the $((I - U^{(t)}A)(\Delta_g x^{(t)}))_i$ should have the same sign with $\text{sign}(x^{(t+1)})$, i.e.

$$
\begin{aligned}
\Delta_g x_i^{(t+1)} &= \text{sign}(x^{(t+1)})(|((I - U^{(t)}A)(\Delta_g x^{(t)}))_i| - |(1 - \mu_{t+1})b^{(t)}|), \\
|\Delta_g x_i^{(t+1)}| &\leq |((I - U^{(t)}A)(\Delta_g x^{(t)}))_i| - (1 - \mu_{t+1})b^{(t)}.
\end{aligned}
\tag{75}
$$

Accumulate all the $|\Delta_g x_i^{(t+1)}|$ with $i \in \text{supp}(x^{(t+1)}) \cup \text{supp}(x_s)$, there is

$$
\begin{aligned}
\|\Delta_g x^{(t+1)}\|_1 &= \sum_{i \in \mathcal{S}^{(t+1)}, i \in \mathbb{S}^{(t+1)}, i \in \{\mathcal{S} - \mathbb{S}^{(t+1)}\}} |\Delta_g x_i^{(t+1)}| \\
&\leq \sum_{i \in \text{supp}(x^{(t+1)})} |((I - U^{(t)}A)(\Delta_g x^{(t)}))_i| + ((|\mathbb{S}^{(t+1)}| - |\mathcal{S}^{(t+1)}|)(1 - \mu_{t+1}) \\
&\quad + (|\mathcal{S}| - |\mathbb{S}^{(t+1)}|))b^{(t)} \\
&\leq (|\mathcal{S}| + |\mathcal{S}^{(t+1)}|)\mu(A)\|\Delta_g x^{(t)}\|_1 + ((|\mathbb{S}^{(t+1)}| - |\mathcal{S}^{(t+1)}|)(1 - \mu_{t+1}) \\
&\quad + (|\mathcal{S}| - |\mathbb{S}^{(t+1)}|))b^{(t)}.
\end{aligned}
\tag{76}
$$

Substitute the $b^{(t)} = \omega_{t+1}(k_{t+1}|\Theta)\mu(A)\sup_{x_s}\|\Delta_g x^{(t)}\|_1$ into Eq. (76). Take the supremum of the right part of Eq. (76):

$$
\begin{aligned}
\|\Delta_g x^{(t+1)}\|_1 &\leq (|\mathcal{S}| + |\mathcal{S}^{(t+1)}|)\mu(A)\|\sup_{x_s}\Delta_g x^{(t)}\|_1 + ((|\mathbb{S}^{(t+1)}| - |\mathcal{S}^{(t+1)}|)(1 - \mu_{t+1}) \\
&\quad + (|\mathcal{S}| - |\mathbb{S}^{(t+1)}|))\omega_{t+1}(k_{t+1}|\Theta)\mu(A)\sup_{x_s}\|\Delta_g x^{(t)}\|_1 \\
&\leq (|\mathcal{S}| + |\mathcal{S}^{(t+1)}| + ((|\mathbb{S}^{(t+1)}| - |\mathcal{S}^{(t+1)}|)(1 - \mu_{t+1}) \\
&\quad + (|\mathcal{S}| - |\mathbb{S}^{(t+1)}|))\omega_{t+1}(k_{t+1}|\Theta))\mu(A)\sup_{x_s}\|\Delta_g x^{(t)}\|_1.
\end{aligned}
\tag{77}
$$

Let

$$
\begin{aligned}
c'_{t+1} &= \sup_{x_s} \log((|\mathcal{S}| + |\mathcal{S}^{(t+1)}| + ((|\mathbb{S}^{(t+1)}| - |\mathcal{S}^{(t+1)}|)(1 - \mu_{t+1}) \\
&\quad + (|\mathcal{S}| - |\mathbb{S}^{(t+1)}|))\omega_{t+1}(k_{t+1}|\Theta))\mu(A)).
\end{aligned}
\tag{78}
$$

Take the supremum of the left part of Eq. (77), the $\sup_{x_s} \|\Delta_g x^{(t+1)}\|_1$ satisfies

$$
\begin{aligned}
\sup_{x_s} \|\Delta_g x^{(t+1)}\|_1 &\leq \exp(c'_{t+1}) \sup_{x_s} \|\Delta_g x^{(t)}\|_1 \\
&\leq \exp(\sum_{i=1}^{t+1} c'_i) \sup_{x_s} \|\Delta_g x^{(0)}\|_1 \\
&\leq \exp(\sum_{i=1}^{t+1} c'_i) sB.
\end{aligned}
\tag{79}
$$

After the upper bound of $\|\Delta_g x^{(t)}\|_1$ is deduced, we shall consider about the relationship between $x^{(t+1)} - x_s$ and $\Delta_g x^{(t)}$:

If $i \in \mathcal{S}^{(t+1)}$, $(x_s)_i = 0$, and $x_i^{(t+1)} \neq 0$. The $i$-th equation in Eq. (69) is

$$
x_i^{(t+1)} - (x_s)_i = ((I - U^{(t)}A)(\Delta_g x^{(t)}))_i - b^{(t)} \text{sign}(x_i^{(t+1)}).
\tag{80}
$$

According to the similar analyses in previous, the sign of $((I - U^{(t)}A)(\Delta_g x^{(t)}))_i$ is the same as $\text{sign}(x_i^{(t+1)})$, $x_i^{(t+1)} - (x_s)_i$ satisfies

$$
\begin{aligned}
x_i^{(t+1)} - (x_s)_i &= x_i^{(t+1)} = (|((I - U^{(t)}A)(\Delta_g x^{(t)}))_i| - b^{(t)}) \text{sign}(x_i^{(t+1)}), \\
|x_i^{(t+1)} - (x_s)_i| &= |x_i^{(t+1)}| = |((I - U^{(t)}A)(\Delta_g x^{(t)}))_i| - b^{(t)}.
\end{aligned}
\tag{81}
$$

If $i \in \mathcal{S}$, the $x_i^{(t+1)} - (x_s)_i$ satisfies

$$
|x_i^{(t+1)} - (x_s)_i| \leq |((I - U^{(t)}A)(\Delta_g x^{(t)}))_i| + b^{(t)}.
\tag{82}
$$

Accumulate all the $|x_i^{(t+1)} - (x_s)_i|$ with $i \in \text{supp}(x^{(t+1)}) \cup \text{supp}(x_s)$, there is

$$
\begin{aligned}
\|x^{(t+1)} - x_s\|_1 &\leq \sum_{i \in \text{supp}(x^{(t+1)})} |((I - U^{(t)}A)(\Delta_g x^{(t)}))_i| + (|\mathcal{S}| - |\mathcal{S}^{(t+1)}|)|b^{(t)}| \\
&\leq (|\mathcal{S}| + |\mathcal{S}^{(t+1)}|)\mu(A)\|\Delta_g x^{(t)}\|_1 + (|\mathcal{S}| - |\mathcal{S}^{(t+1)}|)b^{(t)}.
\end{aligned}
\tag{83}
$$

Substitute the $b^{(t)} = \omega_{t+1}(k_{t+1}|\Theta)\mu(A)\sup_{x_s} \|\Delta_g x^{(t)}\|_1$ into Eq. (83), take the supremum of $\|\Delta_g x^{(t)}\|_1$:

$$
\begin{aligned}
\|x^{(t+1)} - x_s\|_1 &\leq (|\mathcal{S}| + |\mathcal{S}^{(t+1)}|)\mu(A) \sup_{x_s} \|\Delta_g x^{(t)}\|_1 \\
&\quad + (|\mathcal{S}| - |\mathcal{S}^{(t+1)}|)\omega_{t+1}(k_{t+1}|\Theta) \sup_{x_s} \|\Delta_g x^{(t)}\|_1 \\
&\leq (|\mathcal{S}| + |\mathcal{S}^{(t+1)}| + (|\mathcal{S}| - |\mathcal{S}^{(t+1)}|)\omega_{t+1}(k_{t+1}|\Theta))\mu(A) \sup_{x_s} \|\Delta_g x^{(t)}\|_1.
\end{aligned}
\tag{84}
$$

Let $c'^*$ be

$$
c'^* = \log((|\mathcal{S}| + |\mathcal{S}^{(t)}| + (|\mathcal{S}| - |\mathcal{S}^{(t)}|)\omega_t(k_t|\Theta))\mu(A)).
\tag{85}
$$

Substitute Eq. (79) and (85) into Eq. (84), the $l_2$ error bound of LISTA with gain gate should be

$$
\begin{aligned}
\|x^{(t)} - x_s\|_2 \leq \|x^{(t)} - x_s\|_1 &\leq \exp(c'^*) \sup_{x_s} \|\Delta_g x^{(t-1)}\|_1 \\
&\leq \exp(\sum_{i=1}^{t-1} c'_i + c'^*) sB.
\end{aligned}
\tag{86}
$$

Then we shall discuss the value of $c'_i$ and $c'^*$. Let $t_0 = \lceil \log(\frac{sB}{\sigma}) / \log(\frac{1}{(2s-1)\mu(A)}) \rceil$.

When $i \leq t_0$, as $\mu_i = 0$, which means the gain gate does not exist, there is

$$
\begin{aligned}
c'_i &= \sup_{x_s} \log((|\mathcal{S}| + |\mathcal{S}^{(i)}| + (|\mathcal{S}| - |\mathcal{S}^{(i)}|)\omega_{i+1}(k_{i+1}|\Theta))\mu(A)) \\
&= \log((s + k_{i+1} + (s - k_{i+1})\omega_{i+1}(k_{i+1}|\Theta))\mu(A)).
\end{aligned}
\tag{87}
$$

$\exists k_0^i$, s.t. $0 < k_0^i < s$ and $\omega_i(k_0^i|\Theta) < 1 - 1/(s - k_0^i)$. According to main process in the proof Theorem 3, let $k_i = k_0^i$, $0 < k_i < s$, and $c_i' = c_i^* < \log((2s-1)\mu(A))$.

When $i > t_0$, $\sup_{x_s} \|x^{(i)} - x_s\|_1 < sB\exp(ci) \leq \sigma \leq \min_{i \in \text{supp}(x_s)} |(x_s)_i|$. As the minimal absolute value of $x_s$ is less or equal than $\sigma$, $\mathbb{S}^{(t)} = \mathcal{S}$. Select the $b^{(i)}$ so that $k_i = 0$, $\omega_i(k_i|\Theta) \leq 1$. Recall the form in Eq. (78), As $\sup_{x_s} |\mathcal{S}^i| = k^i = 0$, and $c_i'$ is

$$c_i' = \sup_{x_s} \log((|\mathcal{S}| + |\mathcal{S}^{(i)}| + (|\mathcal{S}| - |\mathcal{S}^{(i)}|)\omega_i(k_i|\Theta)\mu(A)) \tag{88}$$
$$\leq \log((s + s(1 - \mu_i))\mu(A)).$$

As $1 - \omega_i(s|\Theta) < \mu_i \leq 1$

$$\begin{aligned}
c_i' &= \log((s + s(1 - \mu_i))\mu(A)) \\
&< \log((s + s\omega_i(s|\Theta))\mu(A)) \\
&< \log((s + s\omega_i(k_0^i|\Theta))\mu(A)) \\
&< \log((s + k_0^i + (s - k_0^i)\omega_i(k_0^i|\Theta))\mu(A)) = c_i^*,
\end{aligned} \tag{89}$$

i.e., $c_i' < c_i^*$. As $\omega_i(\cdot|\Theta)$ is the monotone decreasing function, the second "$<$" in Eq. (89)holds since $k_0^i < s$ and the last "$<$" holds since $k_0^i > 0$.

$|\mathcal{S}| \leq s$, and $|\mathcal{S}^{(t)}| \leq k_t$. According to the assumption of $\omega_t(\cdot|\Theta)$, $\exists k_0^i$, s.t. $0 < k_0^i < s$ and $\omega_i(k_0^i|\Theta) < 1 - 1/(s - k_0^i)$. Select $b^{(t)}$ to let $k_t = k_0^t$. According to the similar derivation in Theorem 3, $c'^*$ in Eq. (85) should satisfies $c'^* \leq \log((s + k_t + (s - k_t)\omega_t(k_t|\Theta))\mu(A)) < \log((2s-1)\mu(A))$. All of the conclusions in Theorem 4 have been proven.

$\square$

## 6.6 PROOF OF PROPOSITION 2

Recall that the update rule of ISTA is $x^{(0)} = \mathbf{0}$ and

$$x^{(t+1)} = s_{\lambda/\gamma}(x^{(t)} - \nabla_x f(x^{(t)})/\gamma). \tag{90}$$

We have the following theorem which analyzes the update rule of ISTA and

$$\eta^* := \arg\min_\eta f(\eta(x^{(t+1)} - x^{(t)}) + x^{(t)}, y) + \lambda\|\eta(x^{(t+1)} - x^{(t)}) + x^{(t)}\|_1. \tag{91}$$

*Proof.* According to the analysis in Section 2 in the main paper, $x^{(t+1)}$ is the solution of minimizing the upper bound $U(x)$,

$$U(x) := f(x^{(t)}, y) + (x - x^{(t)})\nabla_x f(x^{(t)}) + \frac{\gamma}{2}\|x - x^{(t)}\|^2 + \lambda r(x). \tag{92}$$

The sub-gradient of $U(x)$ is

$$\partial_x U(x) = \nabla_x f(x^{(t)}) + \gamma(x - x^{(t)}) + \lambda\partial_x r(x). \tag{93}$$

As the $x^{(t+1)}$ is the optimal solution to minimizing Eq. (92), $\partial_x U(x^{(t+1)})$ satisfies

$$0 \in \partial_x U(x^{(t+1)}) = \nabla_x f(x^{(t)}) + \gamma(x^{(t+1)} - x^{(t)}) + \lambda\partial_x r(x^{(t+1)}), \tag{94}$$

where $r(x) = \|x\|_1$. According to the definition of the sub-gradient, $(\partial_x r(x))_i \in [-1, 1]$ when $x_i = 0$, $(\partial_x r(x))_i = -1$ when $x_i < 0$, and $(\partial_x r(x))_i = 1$ when $x_i > 0$.

From the Eq. (94), there must exists $r_1 \in r(x)$ such that

$$\nabla_x f(x^{(t)}) + \gamma(x^{(t+1)} - x^{(t)}) + \lambda r_1 = 0, \tag{95}$$

where $(r_1)_i = 1$ if $x_i^{(t+1)} > 0$, $(r_1)_i = -1$ if $x_i^{(t+1)} < 0$, and $-1 \leq (r_1)_i \leq 1$ if $x_i^{(t+1)} < 0$.

According to the definition of $\eta^*$ in Eq. (91), we define a new function $\theta(\eta)$ as

$$\theta(\eta) = f(\eta(x^{(t+1)} - x^{(t)}) + x^{(t)}, y) + \lambda\|\eta(x^{(t+1)} - x^{(t)}) + x^{(t)}\|_1. \tag{96}$$

Notice that $\theta(\eta)$ is the line search function of $f(x, y) + \lambda\|x\|_1$. According to the law of convex optimization, as $f(x, y) + \lambda\|x\|_1$ is a convex function, the $\theta(\eta)$ must be also a convex function about $\eta$. The sub-gradient of $\theta(\eta)$ is

$$\begin{aligned}
\partial_x \theta(\eta) = (x^{(t+1)} - x^{(t)})^T \nabla_x f(\eta(x^{(t+1)} - x^{(t)}) + x^{(t)}) + \\
\lambda(x^{(t+1)} - x^{(t)})^T \partial_x r(\eta(x^{(t+1)} - x^{(t)}) + x^{(t)}).
\end{aligned} \tag{97}$$

The $\eta^*$ actually is the value to minimize $\theta(\eta)$ in Eq. (96). There must be

$$0 \in \partial_x \theta(\eta^*). \tag{98}$$

From Eq. (97), the sub-gradient function of $\theta(\eta)$ when $\eta = 1$ is

$$\begin{aligned}
&\partial_\eta \theta(1) \\
=&(x^{(t+1)} - x^{(t)})^T (\nabla_x f(x^{(t+1)}) + \lambda \partial_x r(x^{(t+1)})) \\
=&(x^{(t+1)} - x^{(t)})^T (\nabla_x f(x^{(t+1)}) - \nabla_x f(x^{(t)}) + \nabla_x f(x^{(t)})) + \lambda \partial_x r(x^{(t+1)})) \\
=&(x^{(t+1)} - x^{(t)})^T (\nabla_x^2 f(\zeta)(x^{(t+1)} - x^{(t)}) + \nabla_x f(x^{(t)}) + \lambda \partial_x r(x^{(t+1)})),
\end{aligned} \tag{99}$$

where the last equation holds for Lagrange's mean value theorem and $\zeta \in \mathbb{R}^n$. Substitute $\nabla_x f(x^{(t)})$ in Eq. (95) into Eq. (99), the $\partial_\eta \theta(1)$ is

$$\begin{aligned}
&\partial_\eta \theta(1) \\
=&(x^{(t+1)} - x^{(t)})^T (\nabla_x^2 f(\zeta)(x^{(t+1)} - x^{(t)}) - \gamma(x^{(t+1)} - x^{(t)}) - \lambda r_1 + \lambda \partial_x r(x^{(t+1)})) \\
=&(x^{(t+1)} - x^{(t)})^T ((\nabla_x^2 f(\zeta) - \gamma I)(x^{(t+1)} - x^{(t)}) + \lambda(\partial r(x^{(t+1)}) - r_1)) \\
=&(x^{(t+1)} - x^{(t)})^T (\nabla_x^2 f(\zeta) - \gamma I)(x^{(t+1)} - x^{(t)}) + \\
&\lambda \sum_i (x_i^{(t+1)} - x_i^{(t)})((\partial r(x^{(t+1)}))_i - (r_1)_i),
\end{aligned} \tag{100}$$

where the sub-gradient $\partial r(x^{(t+1)})$ is a set. For all $r^* \in \partial r(x^{(t+1)})$,

$$\begin{aligned}
&(x^{(t+1)} - x^{(t)})^T (\nabla_x^2 f(\zeta) - \gamma I)(x^{(t+1)} - x^{(t)}) + \\
&\lambda \sum_i (x_i^{(t+1)} - x_i^{(t)})((r^*)_i - (r_1)_i) \in \partial_\eta \theta(1).
\end{aligned} \tag{101}$$

We prove $\eta^* \geq 1$ according to the counter-evidence. According to the properties of convex function and the sub-gradient, assume $\eta^* < 1$, s.t. $0 \in \partial_\eta \theta(\eta^*), \forall r_\theta \in \partial_\eta \theta(1)$, there will be

$$r_\theta > 0.$$

However, as $r_1 \in \partial r(x^{(t+1)})$, substitute $r = r_1 \in \partial r(x^{(t+1)})$ into Eq. (100). The corresponding element in sub-gradient when $r = r_1$ is $r_\theta = (x^{(t+1)} - x^{(t)})^T (\nabla_x^2 f(\zeta) - \gamma I)(x^{(t+1)} - x^{(t)}) \in \partial_\eta \theta(1)$. According to given condition $\gamma I - \nabla_x^2 f(x) \succ 0$, $r_\theta < 0$, which is in contrast to $r_\theta > 0$. Therefore, the conclusion $\eta^* \geq 1$ is obtained.

Moreover, consider about the last term of Eq. (100), i.e.

$$\sum_i (x_i^{(t+1)} - x_i^{(t)})((\partial r(x^{(t+1)}))_i - (r_1)_i). \tag{102}$$

If $\text{supp}(x^{(t)}) \subset \text{supp}(x^{(t+1)})$, there are two situations about index $i$. 1) $i \in \text{supp}(x^{(t+1)})$, there will be $x_i^{(t+1)} \neq 0$ and $(\partial r(x^{(t+1)}))_i = (r_1)_i = \text{sign}(x_i^{(t+1)})$. 2) $i \notin \text{supp}(x^{(t+1)})$ and $i \notin \text{supp}(x^{(t)})$, there will be $x_i^{(t+1)} = x_i^{(t)} = 0$. Both conditions will make the term $(x_i^{(t+1)} - x_i^{(t)})((\partial r(x^{(t+1)}))_i - (r_1)_i)$ in Eq. (102) be 0. Therefore, Eq. (102) is

$$\sum_i (x_i^{(t+1)} - x_i^{(t)})((\partial r(x^{(t+1)}))_i - (r_1)_i) = 0. \tag{103}$$

According to the given condition $\gamma I - \nabla_x^2 f(x) \succ 0$, $\partial_\eta \theta(1)$ should be a number but not a set and $\partial_\eta \theta(1) = (x^{(t+1)} - x^{(t)})^T (\nabla_x^2 f(\zeta) - \gamma I)(x^{(t+1)} - x^{(t)}) < 0$. As the $\theta(\eta)$ is convex function, there must be $\eta^* > 1$ because of $0 \in \partial_\eta \theta(\eta^*)$. The conclusion $\eta^* > 1$ is derived. $\qquad \square$

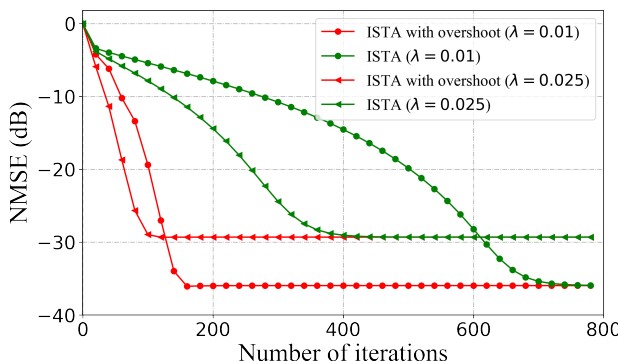

Figure 7: Experimental results validating our Proposition 2. It can be observed that the update of ISTA "lags behind".

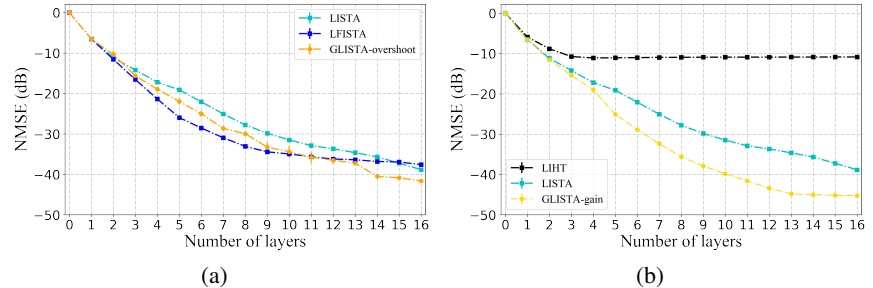

Figure 8: Comparison of overshoot and gain gate with similar methods over five runs.

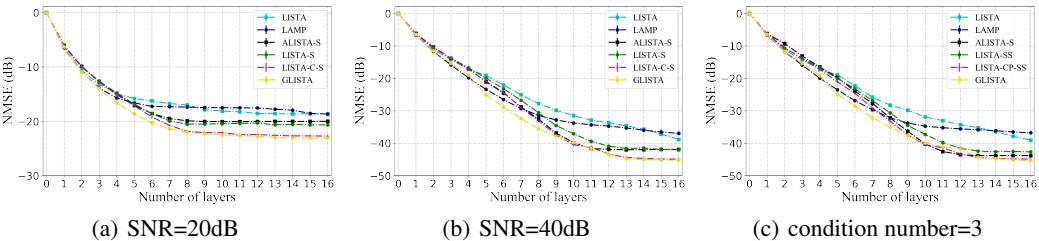

(a) SNR=20dB        (b) SNR=40dB        (c) condition number=3

Figure 9: Comparison of sparse coding methods under different settings over five runs. Our GLISTA consistently outperforms the competitors in almost all test cases with different numbers of layers.

## 7 MORE SIMULATION EXPERIMENTS

**Validation of Proposition 2**: Some more experimental results are given here due to the length limit of the main body of our paper. One might also be interested in our Proposition 2, hence we first conduct an experiment to confirm it. We adopt ISTA with an adaptive overshoot and compare it with the standard ISTA for sparse coding. The adaptation is obtained via enlarging the step size from 1.0 through backtracking line search (see section 7 for more details). Figure 7 demonstrates that our overshoot mechanism facilitates ISTA optimization, and such a result confirms Proposition 2.

**Comparison with similar methods**: As mentioned in the main body of the paper, the overshoot gates is proposed do address insufficient step size, which is similar to the motivation of (L)FISTA. LIHT and support select can also be considered as special cases of our gain gates (by letting $\mu_t = 1$ in the inverse proportional function). We compare these similar methods with our overshoot and gain gates in Figure 8. It can be seen that when compared with LISTA, LFISTA converges faster in lower layers, and our overshoot gates also show such advantage. When applying to deeper layers, LFISTA converges quite slow while the overshoot gates still perform well, which indicates that the time-varying property is beneficial in practice. LISTA with our gain gates is obviously better than LIHT as shown in Figure 8(b), and sufficient experimental results in the paper also prove that the gain gate outperforms support select (e.g., in LISTA-C-S and LISTA-S).

**Comparison under less challenging settings**: Now we also give sparse coding results under the described less challenging settings on the noise level and the condition number in Figure 9. Compared with LISTA-CP, LAMP, LISTA-SS, and LISTA-CP-SS, our gated LISTA (GLISTA) performs remarkably better with less ill-posed dictionary matrices and less noises. Table 3 and 4 report the statistical means and standard deviations of five runs using different methods. It can be seen that the improvement achieved by our GLISTA is significant.

Table 3: Comparison of the final NMSEs under different noise levels with $d = 16$. The condition number of the dictionary is not specifically constrained.

| SNR | LISTA | LAMP | LISTA-S | LISTA-C-S | ALISTA-S | GLISTA (ours) |
|---|---|---|---|---|---|---|
| 40 | -38.72±0.09 | -36.77±0.60 | -41.99±0.09 | -44.85±0.02 | -41.86±0.04 | **-45.22±0.02** |
| 20 | -18.65±0.09 | -18.66±0.09 | -20.64±0.06 | -22.84±0.02 | -20.00±0.05 | **-23.08±0.03** |
| 10 | -9.42±0.08 | -9.46±0.66 | -9.84±0.02 | -11.06±0.01 | -9.04±0.02 | **-11.41±0.02** |

Table 4: Comparison of the final NMSEs under different condition numbers with $d = 16$. The noise level is chosen as SNR=40dB for all the tested condition numbers.

| Con. num. | LISTA | LAMP | LISTA-S | LISTA-C-S | ALISTA-S | GLISTA (ours) |
|---|---|---|---|---|---|---|
| 3 | -39.03±0.54 | -37.26±0.13 | -43.12±0.06 | -44.90±0.03 | -43.88±0.26 | **-45.33±0.04** |
| 30 | -29.65±0.89 | -28.44±0.31 | -32.30±0.17 | -38.36±0.57 | -31.50±0.15 | **-39.61±0.64** |
| 100 | -21.39±0.75 | -22.23±0.18 | -27.08±0.57 | -27.94±0.34 | -27.10±0.02 | **-34.07±0.64** |

## 8 PROGRESSIVE TRAINING AND ADAPTIVE OVERSHOOT

Our training mostly follows it of Chen et al.'s (2018), and some key steps are listed here: 1) The model is trained progressively to include more layers during the training phase. At the very beginning, only learnable parameters in the first layer is considered, and parameters in the second layer is only included once training on the first update converges, so as the third and higher layers. 2) Training after including the $t$-th layer is split into three stages, with an initial learning rate of $0.0005$ to optimize its own learnable parameters first, and learning rates of $0.0001$ and $0.00001$ to jointly optimize all learnable parameters from the $0$-th to $t$-th layers in the second and third stages, respectively. We move to the next stage once no performance gain is observed on the validation set for 4000 iterations. 3) With the three stages done on the $t$-th layer, training moves to include the $(t + 1)$-th and the same three stages of training are performed.

We perform an adaptive overshoot in the experiment to confirm Proposition 2. The algorithm is summarized in Algorithm 1. Most of input variables are introduced in the main body of our paper and $\tau$ is given as the step size for performing line search. The whole algorithm procedure is very similar to the famous backtracking line search. The step size $\eta$ for sparse coding is updated by $\tau$ until the objective function $f(x, y) + \lambda r(x)$ does not decrease any more.

---

**Algorithm 1** ISTA with adaptive overshoot.

---

**Input:** The dictionary matrix $A$, an observation $y$, an initial step size $\eta_0 = 1.0$ for sparse coding, a step size $\tau = 1.05$ for line search, and a maximal number of iteration.
**Output:** output result
1: $x^{(0)} = \mathbf{0}$;
2: **for** $t = 0, \cdots, K - 1$ **do**
3:      $\tilde{x}^{(t)} = s_{\lambda/\gamma}((I - A^T A/\gamma)x^{(t-1)} + A^T y/\gamma)$;
4:      $x_p = \tilde{x}^{(t)}, \eta = \eta_0 \tau$;
5:      $x_c = \tau(\tilde{x}^{(t)} - x^{(t)}) - x^{(t)}$;
6:      **while** $f_o(x_p, y) \geq f_o(x_c, y)$ **do**
7:          $x_p = x_c, \eta = \tau\eta$;
8:          $x_c = \eta(\tilde{x}^{(t)} - x^{(t)}) - x^{(t)}$;
9:      $x^{(t)} = x_p$;
10: **return** $x^{(K-1)}$

---

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
