# OpenReview forum: "Sparse Coding with Gated Learned ISTA"
_ICLR.cc/2020/Conference — Accept (Spotlight)_

### Official Review · AnonReviewer2 · 2019-10-22
**Official Blind Review #2**

**Rating:** 8

**Review:**

1. Summary
The authors propose extensions to LISTA with the goal of addressing underestimation (by introducing “gain gates”) and including momentum (by introducing “overshoot gates”). The authors provide theoretical analysis for each step of their LISTA augmentations, showing that it improves convergence rate. Their theoretical statements are empirically validated and then a numerical comparison is performed between the most interesting LISTA variants. Their proposed GLISTA performs favorably, especially for networks of depth greater than 10.
2. Decision and arguments
The theory given is quite comprehensive and seems solid. Moreover it’s motivated by and helps real problems, namely L1’s well-known underestimation and LISTA’s lack of momentum. On top of that, the numerical results not only show your GLISTA outperforming others, but that by adding gates to other LISTA variants you can get better results. On the other hand (see details in the Questions section), there is some critical information missing from the empirical section which makes the results non-repeatable.
Training is not described, i.e. hyperparameter searches and stopping criteria. A couple of the plots validating theory are difficult to understand. The datasets used are synthesized and tiny, only 1000 samples (training/testing/validation sets are not described) and 250x500 dimensions. That means your network has many more parameters than data samples unless I am mistaken.
So I have given weak reject because the paper has strong theory but weak experiments making it hard to trust the conclusions. I really want to hear back about the questions raised below.
3. Questions
a) Just before eqn 12 you say that gains greater than 1 can be more appropriate. Am I understanding this correctly: *before* shrinkage you want to apply a gain on code elements that are “truly” nonzero, in order to cancel out the imminent L1 penalization? And you will learn to predict those code elements via parameters Lambda?
b) Just before Section 4, you describe the difference between yours and Moreau & Bruna’s momentum taps. If I understand correctly they have a matrix called W_m^(k) which is multiplied onto the previous iterate—but for each layer / time unit (k), the matrix may vary by learning (note that in the main body of their paper they don’t explicitly use ^(k) notation, but it is explained to be the same as the other LISTA parameter matrices which vary with time, and is made explicit in the appendix). So I think their momentum is in fact time-varying. Moreover it is certainly dependent on the previous iterate, using the function f(z^(k-1)) = W_m^(k) * z^(k-1). Did you mean something else by that? In what sense do you have a higher capacity—do you have more parameters?
c) If I understand correctly, in Figure 5a, 5b, for each point you have trained independently an N-layer GLISTA. Then you observe properties of your learned parameters.
For 5a: how do you compute the “output” of the gain gates? What is the input that gives this output?
For 5b: which ‘t’ is used to calculate ||W(t)D-I+U(t)A||? Or did you just train a single 15-layer GLISTA, and the x-axis is just ‘t’?
d) Could you tell us more about the hyperparameter tests for every method? How do we know it was a fair comparison? There are no error bars, but from experience I know that training LISTA type networks can be a pain. What algorithms did you use? Stopping criteria? Some plots start at zero layers (4c, 5b, 6a-c) and some start at 1 layer…
e) Interesting that in your real-data example, the sparse vector e has more non-zeros than zeros, is that really sparse? What dictionary A did you use, then? By your own description this task does not fit the sparse coding model you have analyzed…. Am I misunderstanding?
4. Additional feedback/ minor comments
a) Add reference for DOA estimation application
b) You should put something like “we prove this in the supplement” for props and theorems. Otherwise to readers less familiar with the literature it will seem like you forgot to put a reference.
c) It seems unnecessary to put ||epsilon||<= 0 , instead of epsilon=0.
d) Basically I think you should explain what your gate is before providing proofs about it. It would be useful to see a plot of the function g and/or kappa so provide some intuition about what you are doing to the architecture (nonlinear? Linear? Threshold? Etc.). Capital “Lambda” is not defined until after theorems are provided about the functions g, kappa, f. In fact it seems like kappa is only useful for the sake of proofs. The first time through, I over-read the statement “All the learnable parameters are thus collected as Lambda…” just before Section 3.1.1 because I was thinking “Why isn’t Lambda in the definitions of f/g/kappa??”. Anyway it makes perfect sense when you define it clearly as after Eqn 18.
e) At the beginning of Section 3.2 you have a clause “the over-shoot gates act more like on the output”, which doesn’t make sense.
f) Figures should be approximately self-explanatory—but “Ratio of lower components” is not explained in the caption of Fig 4b. Although I greatly appreciate that you have made the actual plot lines/markers very easy to see! Should these results be averaged over many training attempts with error bars?


**Experience Assessment:**

I have published one or two papers in this area.

**Review Assessment: Checking Correctness Of Derivations And Theory:**

I assessed the sensibility of the derivations and theory.

**Review Assessment: Checking Correctness Of Experiments:**

I carefully checked the experiments.

**Review Assessment: Thoroughness In Paper Reading:**

I read the paper thoroughly.

---

> ### Author Response · Authors · 2019-11-12
> **Response to AnonReviewer #2**
>
> We appreciate the positive feedback on our theoretical discoveries and suggestions about clarifying more implementation details. We have revised the paper (especially Section 4) to introduce our experimental settings clearer, including but not limited to the objective function, training policy, stoping criteria, optimizer, and hyper-parameter selection. Our training strategies are kept the same as prior works for fair comparisons, following a progressive training principle on streaming data [1][2][3]. In principle, the training set “grows” as training proceeds and is by no means tiny. The test set and validation set each consist of 1000 samples and the same sets are used for evaluating different methods. Our code will be made publicly available to make the results more reproducible.
>
> Responses to majors concerns and questions:
>
> a. The development of our gain gains is motivated by rigorous analyses in Proposition 1. Since we know that, for non-zero code elements, the absolute value of estimations are always smaller or at most equal to that of the ground-truth and they share the same signs, we aim to enlarge such estimations before feeding them into the next processing units. Indeed, the estimations are enlarged via learnable gate functions in our method.
>
> b. We are aware of the ^(k) notation in the appendix of LFISTA, but what we tried to explain right before Section 4 is that, the major difference between our method and some classical momentum-based methods is whether the scaling factors (multiplied on previous estimations) in the momentum taps are dependent on the current input(s). Note that the “current input(s)” here indicate not only the previous estimation x^(t-1), but also the observation y. We have revised the content a bit in the paper for better clarity on this point.
>
> c. Actually, unlike in Figure 4c, 6, and 7 we train a single 16-layer network for Figure 5a/5b and observe properties on each of its intermediate/final estimations. Therefore, for the output of the gain gates, we just average the obtained gain values over the whole test set and calculate it on all intermediate/final layers (from t=0 to t=d-1=15) for plotting in Figure 5a. Figure 5b is similarly obtained. We have revised the figure caption to make it clearer.
>
> d. 1) We have revised Section 4 in the paper to add more training details and avoid possible confusion. Hyper-parameters in competitors are set/tuned following their official codes on the validation set, and similarly for the ones in our method. 2) We now report standard derivations as well in Table 1, 3, and 4, along with the average performance over 5 runs which has already been reported in our previous version. All the figures except for Figure 5 now illustrate average results along with error bars as suggested. 3) Some plots start at the “first” layer instead of the “zeroth” layer since some estimations may not make much sense with an initial estimation, e.g., for "false positive rate" within non-zero code components shown in Figure 4a, we don’t calculate it on x^(0)=0.
>
> e. There seems to be a typo in our Section 4.2. Actually we let 40% of the elements of e be non-zero (rather than be “zero” as described in the previous version). We also tried with higher sparsity with only 20% of the elements being non-zero and our method still outperforms the others
> (ours: 5.19e-3$\pm$0.53e-3, LISTA:  0.0402$\pm$0.0050, if q=15;
>  ours: 1.76e-3$\pm$0.84e-3, LISTA:   0.0204$\pm$0.0032, if q=25;
>  ours: 3.79e-4$\pm$0.39e-4, LISTA:  7.51e-3$\pm$2.97e3, if q=35).
> As suggested, we also give a clearer description of the photometric stereo analysis problem in Section 4.2, including its corresponding A and y, to show that the problem well fits the model we have analyzed.
>
> For minor comments, we thank the reviewer for pointing out and we have revised the paper accordingly.
>
> [1] Chen X, Liu J, Wang Z, et al. Theoretical linear convergence of unfolded ISTA and its practical weights and thresholds. NeurIPS, 2018.
>
> [2] Liu J, Chen X, Wang Z, et al. ALISTA: Analytic weights are as good as learned weights in LISTA, ICLR, 2019.
>
> [3] Borgerding M, Schniter P, Rangan S. AMP-inspired deep networks for sparse linear inverse problems. IEEE Transactions on Signal Processing, 2017, 65(16): 4293-4308.

---

### Official Review · AnonReviewer1 · 2019-10-23
**Official Blind Review #1**

**Rating:** 8

**Review:**

Summary:

This paper is focused on solving sparse coding problems using LISTA-type
networks. It discusses the weakness of the ``no false positive'' assumption in
previous works and the weakness results in underestimated code components. The
authors propose a ``gain gating function'' to mitigate the weakness. Moreover,
the paper incorporates another ``overshoot'' gating function inspired by
momentum-based methods. Both contributions are supported with theoretical and
empirical results. Numerical experiments show that the proposed model is
superior to previous works especially in cases with high measurement noises or
ill-composed basis matrix. The paper is well written and easy to follow, and the
empirical results are impressive.

I really like the relaxation of ``no false positive'' assumption as in real world
application, learning-based algorithms may find it difficult to keep satisfying
the assumption while maintaining good empirical performance. And this relaxation
contributes to guarantee the convergence in a more real scenario.

I think in (Chen et al., 2018), the support selection technique serves to
mitigate the problem of underestimated code components as it bypasses the
thresholding function for codes with large magnitudes. The empirical results
also show that in many cases LISTA with support selection has comparable
performance. I don't know if LISTA with support selection also suffers from
this underestimation problem severely.

Questions:

1. The theorems in Section 3.1 hold without overshooting mechanism, i.e. eta==1.
Prop. 2 says that for one step of iteration, the optimal updates requires
overshoot. When the gain and overshoot mechanisms are combined, however, will
the theoretical convergence still hold?

2. I don't have a very good understanding of why GLISTA performs so well given
ill-composed basis matrix compared to previous works. Could there be some
(intuitive) explanation as to this based on the theoretical results?

3. More details about the training process should be included. For example, I
can guess the loss function used for the training. But it is confusing not
mentioning it at all in the paper. Some might think the objective funciton in
eqn (2) but the convergence analysis is with respect to the ground truth sparse
vector. Also, the training scheme and some hyperparameter selection should also
be stated at least in Appendix.


Overall I hold very positive attitude towards this paper due to its theoretical
contributions and good empirical results.

================================
Update: the authors addressed my questions well and I will keep my positive decision on this paper.

**Experience Assessment:**

I have published one or two papers in this area.

**Review Assessment: Checking Correctness Of Derivations And Theory:**

I assessed the sensibility of the derivations and theory.

**Review Assessment: Checking Correctness Of Experiments:**

I carefully checked the experiments.

**Review Assessment: Thoroughness In Paper Reading:**

I read the paper thoroughly.

---

> ### Author Response · Authors · 2019-11-13
> **Response to AnonReviewer #1**
>
> Thanks for the positive feedback. We are very excited that our theoretical studies for relaxing the assumption of “no-false-positive” are endorsed. If possible, we would also like to study the support selection technique more to see whether the underestimation is theoretically mitigated in future studies.
>
> Response to your questions:
>
> 1. Good question! We analyzed the scenario when gain gates are introduced and eta>1 (i.e., with constant overshoots) and found that our theoretical results of linear convergence hold as well. For learned overshoot as described in Eq. (17) and function o_t, the theoretical analysis is nontrivial and we would like to consider it carefully in the future.
>
> 2. In our understanding, one difference between ill-posed dictionary matrices and well-posed ones is their different values of $\mu(A)$, and according to the theoretical results, ill-posed $A$ leads to slower convergence and possibly requires more sophisticated gains. We conjecture that our learnable mechanism endows higher capacity to adapt the model to such scenarios. From another perspective, we know from [1] that the convergence of LISTA with noise fulfills $$\|x^{(t)}-x_s\|_2\leq \alpha^t sB +\frac{C^\ast}{1-\alpha},$$ in which $\alpha=(2s-1)\mu(A)$ is large given an ill-posed dictionary matrix and $C^\ast>0$ does not vary with $\mu(A)$. For a deep sparse coding network with noise, the second term in the above bound dominates the convergence speed. Since our gain gates operate as decreasing $\alpha$ (when incorporated into LISTA), its superiority may be more significant with an originally large $\alpha$.
>
> 3. We appreciate the suggestions for including more experimental details. Explanations about our objective function, training policy, stoping criteria, optimizer, and hyper-parameter selection have now been added into Section 4 and 8.
>
> [1] Chen X, Liu J, Wang Z, et al. Theoretical linear convergence of unfolded ISTA and its practical weights and thresholds. NeurIPS, 2018.

---

### Official Review · AnonReviewer3 · 2019-11-04
**Official Blind Review #3**

**Rating:** 8

**Review:**

This paper proposed two novel gates to improve the convergence speed of the learned ISTA (LISTA) algorithm for sparse coding. The first gate is designed to address the problem that the output of the LISTA algorithm usually has a lower magnitude compared against the ground-truth. To address this, the paper proposed a gain gate to increase the magnitude of a layer in the LISTA algorithm. The second gate is designed to further improve the convergence with a technique similar to the momentum but with a time-varying coefficient.

This paper provides rigorous theoretical justifications for the observations that motivate the two gates in two propositions. It also provides theoretical guarantees for the first gain gate under practical assumptions.

Thorough synthetic experiments are conducted to empirically validate the proposals as well as the theorems. The effectiveness of the gates is also proved in a real-world computer vision task, photometric stereo.

Given the thoroughness of the theoretical and experimental justifications, I strongly recommend the acceptance of the paper.

Despite the strength of the paper, I have the following suggestions and questions regarding the two gates proposed by the paper:

1. The second proposed gate, the overshoot gate, lacks theoretical justification and is not as well studied as the gain gate. The experiments for this gate are not included in the main text but got placed in the appendix. I suggest squeezing some of them into the main text to show a complete picture about the overshoot gate. If there are more theoretical results about the behavior of the overshoot gate, the author should include it in the paper; otherwise, the author should clearly states this limitation in the main text of the paper.

2. The author proposed two gates to address two different issues of the original LISTA algorithm. However, the two gates are only applied seperately to modify the LISTA algorithm; they have not been integrated into a single concise algorithm. It would strengthen the paper if the author could provide a further discussion on how the two gates can be combined to address the two issues in one algorithm.


**Experience Assessment:**

I have published one or two papers in this area.

**Review Assessment: Checking Correctness Of Derivations And Theory:**

I did not assess the derivations or theory.

**Review Assessment: Checking Correctness Of Experiments:**

I assessed the sensibility of the experiments.

**Review Assessment: Thoroughness In Paper Reading:**

I made a quick assessment of this paper.

---

> ### Author Response · Authors · 2019-11-13
> **Response to AnonReviewer #3**
>
> Thanks for the positive feedback and constructive suggestions. We have revised the paper accordingly. Some overshoot gate related experimental results have been squeezed into Section 4.1 in the main body of the paper. The effectiveness of a direct combination of the overshoot and gain gates has been highlighted. Also, the progress and limitation of asymptotic behaviors with overshoots are explained carefully in Section 3.2.

---

### Author Response · Authors · 2019-11-15
**General response from authors**

We thank all the reviewers for their time and effort in reading and reviewing our paper. We have revised the paper following constructive suggestions and comments from the reviewers and we are more than glad to answer any further questions. In particular:

* We have squeezed more experiments and discussions for the overshoot and gain gate into the main body of the paper.

* We have shown standard deviations and error bars along with average performance over five runs in empirical comparisons.

* We have explained more about our experimental details in Section 4 and 8 in the paper.

---

### Decision · Program_Chairs · 2019-12-19

**Decision:**

Accept (Spotlight)

**Comment:**

The paper extends LISTA by introducing gain gates and overshoot gates, which respectively address underestimation of code components and compensation of small step size of LISTA. The authors theoretically analyze these extensions and backup the effectiveness of their proposed algorithm with encouraging empirical results. All reviewers are highly positive on the contributions of this paper, and appreciate the rigorous theory which is further supported by convincing experiments. All three reviewers recommended accept.